# The WD40 domain of ATG16L1 is required for its non-canonical role in lipidation of LC3 at single membranes

Katherine Fletcher[1,†], Rachel Ulferts[2,†], Elise Jacquin[1], Talitha Veith[2], Noor Gammoh[3], Julia M Arasteh[4], Ulrike Mayer[5], Simon R Carding[6] (iD), Thomas Wileman[4], Rupert Beale[2,*] (iD) & Oliver Florey[1,**] (iD)

## Abstract

A hallmark of macroautophagy is the covalent lipidation of LC3 and insertion into the double-membrane phagophore, which is driven by the ATG16L1/ATG5-ATG12 complex. In contrast, non-canonical autophagy is a pathway through which LC3 is lipidated and inserted into single membranes, particularly endolysosomal vacuoles during cell engulfment events such as LC3-associated phagocytosis. Factors controlling the targeting of ATG16L1 to phagophores are dispensable for non-canonical autophagy, for which the mechanism of ATG16L1 recruitment is unknown. Here we show that the WD repeat-containing C-terminal domain (WD40 CTD) of ATG16L1 is essential for LC3 recruitment to endolysosomal membranes during non-canonical autophagy, but dispensable for canonical autophagy. Using this strategy to inhibit non-canonical autophagy specifically, we show a reduction of MHC class II antigen presentation in dendritic cells from mice lacking the WD40 CTD. Further, we demonstrate activation of non-canonical autophagy dependent on the WD40 CTD during influenza A virus infection. This suggests dependence on WD40 CTD distinguishes between macroautophagy and non-canonical use of autophagy machinery.

**Keywords** autophagy; phagocytosis; LC3; ATG16L1; influenza
**Subject Categories** Autophagy & Cell Death; Immunology
**The EMBO Journal (2018) 37: e97840**

See also: **D Fracchiolla & S Martens** (February 2017)

## Introduction

Autophagy is a catabolic process where cytosolic components are degraded within the lysosome. Canonical autophagy (here referring to macroautophagy) involves the formation of double-membrane vesicles called autophagosomes that sequester intracellular material, including organelles or proteins, and target it to lysosomes (Feng *et al*, 2013). Canonical autophagy is activated by a variety of cellular stresses such as nutrient deprivation, and functions to maintain cellular energy metabolism and viability (Choi *et al*, 2013).

The association of microtubule-associated protein 1 light chain 3 (LC3) has long been considered a defining hallmark of autophagosomes. LC3 is a cytosolic ubiquitin-like protein, which upon activation of canonical autophagy becomes covalently bound to phosphatidylethanolamine (PE) on autophagosomal membranes (Mizushima *et al*, 1998). It is now appreciated that the membrane remodelling machinery required for starvation-induced autophagy can be co-opted to a variety of different uses (Codogno *et al*, 2011). For example, LC3 can be conjugated to PE in the context of single-membrane, non-autophagosome compartments. We refer to these processes, which target LC3 to a single membrane, as "non-canonical autophagy" (Florey & Overholtzer, 2012). Examples of this non-canonical autophagy pathway include LC3-associated phagocytosis (LAP), where LC3 is lipidated at single-membrane phagosomes following the engulfment of bacterial and fungal pathogens or apoptotic and necrotic cells (Sanjuan *et al*, 2007; Florey *et al*, 2011; Martinez *et al*, 2011). A similar LAP-like LC3 lipidation event is seen during macropinocytosis and entosis, the latter a form of live cell engulfment (Florey *et al*, 2011). It has also recently been reported that a range of drugs possessing lysosomotropic or ionophore properties, including monensin, CCCP and chloroquine, are able to activate non-canonical autophagy and induce the lipidation of LC3 at single-membrane compartments of the endolysosomal

1  Signalling Programme, Babraham Institute, Cambridge, UK
2  Division of Virology, Department of Pathology, University of Cambridge, Cambridge, UK
3  Edinburgh Cancer Research UK Centre, University of Edinburgh, Edinburgh, UK
4  Norwich Medical School, UEA, Norwich, UK
5  School of Biological Sciences, UEA, Norwich, UK
6  Quadrum Institute Bioscience, Norwich Research Park, Norwich, UK
   *Corresponding author. Tel: +44 1223 763421; Fax: +44 1223 336926; E-mail: rclb2@cam.ac.uk
   **Corresponding author. Tel: +44 1223 496431; Fax: +44 1223 496043; E-mail: Oliver.Florey@babraham.ac.uk
   †These authors contributed equally to this work

system (Florey *et al*, 2015; Jacquin *et al*, 2017). While these processes associated with unconventional LC3 lipidation are not *bona fide* autophagic processes (Galluzzi *et al*, 2017), they are commonly referred to as non-canonical autophagy (Henault *et al*, 2012; Kim *et al*, 2013; Cadwell, 2016; Martinez *et al*, 2016; Jacquin *et al*, 2017). This should not be confused with ATG5- or Beclin1-independent autophagy (Codogno *et al*, 2011).

Non-canonical autophagy has been implicated in regulating the degradation of material following macro-scale engulfments through the modulation of lysosome fusion with macroendocytic compartments. This is important for many immune responses such as pathogen clearance and antigen presentation (Sanjuan *et al*, 2007; Ma *et al*, 2012; Romao *et al*, 2013). LAP also plays a role in modulating the cytokine profile in macrophages following the engulfment of apoptotic cells (Martinez *et al*, 2011, 2016) resulting in a proinflammatory response. In the absence of LAP, mice develop an autoimmune phenotype that resembles systemic lupus erythematosus (Martinez *et al*, 2016). The exact molecular mechanisms underlying how non-canonical autophagy facilitates these events remain unknown.

An important feature common to non-canonical autophagy processes is that the associated LC3 lipidation is independent from the upstream regulators of canonical autophagy, including the ULK1 complex containing ATG13 and FIP200, and nutrient status (Florey *et al*, 2011; Martinez *et al*, 2015). However, like canonical autophagy, non-canonical autophagy utilises the core ubiquitin-like conjugation machinery that consists of ATG3, 4, 5, 7, 10, 12, 16L1, which together act together to co-ordinate the lipidation of LC3 with PE.

*In vivo*, ATG16L1 is responsible for the correct targeting of LC3 to forming autophagosome membranes (Fujita *et al*, 2008). It contains an N-terminal domain which associates with ATG5 and ATG12, a central FIP200 and WIPI2b binding domain, and a C-terminal WD40 domain (WD40 CTD). The ATG5-ATG12-ATG16L1 complex has an E3-Ubiquitin ligase-like enzymatic activity required for lipidation of LC3 and is essential for LC3 targeting to membranes during canonical and non-canonical autophagy (Fujita *et al*, 2008; Martinez *et al*, 2015). The central region in ATG16L1 (amino acids 229–242) encompassing binding sites for both FIP200 and WIPI2b is known as the FIP200 binding domain (FBD). Deletion of this domain prevents ATG16L1 recruitment to forming autophagosomes and inhibits the canonical autophagy response to both amino acid starvation and infection by cytosolic bacteria (Gammoh *et al*, 2013; Dooley *et al*, 2014). Recruitment of the ATG16L1 complex to forming autophagosomes is dependent on the generation of PI3P, via the type III phosphatidylinositol 3-kinase VPS34. Accordingly, inhibition of VPS34 with wortmannin and other inhibitors abrogates autophagosome formation and the associated LC3 lipidation (Itakura & Mizushima, 2010). Subsequently, the PI3P binding effector WIPI2b and FIP200, a member of the ULK1 complex, directly bind and recruit the ATG16L1 complex (Gammoh *et al*, 2013; Nishimura *et al*, 2013; Dooley *et al*, 2014). The C-terminal WD40 domain of ATG16L1 is not present in Atg16, the yeast homolog. The structure of the WD40 CTD has recently been solved, but its biological function remains unclear, although there is some evidence that the WD40 CTD can bind ubiquitin and other factors involved in lysophagy and some forms of xenophagy (Fujita *et al*, 2013; Boada-Romero *et al*, 2016; Bajagic *et al*, 2017).

Considering the importance of ATG16L1 in LC3 lipidation during both canonical and non-canonical pathways, we sought to determine the mechanism by which ATG16L1 functions specifically during non-canonical autophagy. In this report, we reveal a critical role for the WD40 CTD of ATG16L1 in its recruitment to single-membrane endolysosomal compartments and for LC3 lipidation during non-canonical autophagy. Importantly, canonical autophagy does not appear to be affected by deletion of the WD40 CTD of ATG16L1. Thus, our results provide the first means to genetically distinguish between canonical and non-canonical autophagy.

Influenza A virus (IAV) infection results in the lipidation of LC3 and its relocalisation to the plasma membrane and to perinuclear structures (Gannage *et al*, 2009; Beale *et al*, 2014). This depends on the viral M2 protein, a proton channel with multiple roles in the viral life cycle. Targeting of LC3 to the plasma membrane is promoted by a direct interaction between a LC3-interacting region (LIR) in the C-terminal tail of M2 and LC3. Here, using our strategy to distinguish canonical and non-canonical autophagy, we demonstrate that LC3 relocalisation during IAV infection depends on the proton channel activity of M2 and the WD40 CTD of ATG16L1, raising the possibility that activation of the non-canonical autophagy pathway can be triggered by loss of cellular pH gradients.

## Results

### ATG16L1 recruitment to membranes and LC3 lipidation during non-canonical autophagy does not require VPS34 and WIPI2b

ATG16L1 in complex with ATG5 and ATG12 acts as an E3 enzyme that lipidates LC3 to PE in membranes. During canonical autophagy, ATG16L1 is responsible for targeting the complex to sites of forming autophagosomes. We reasoned that ATG16L1 may also direct LC3 lipidation during non-canonical autophagy, and to address this, first examined its recruitment. In agreement with published work (Fujita *et al*, 2008), we detect ATG16L1 colocalised with LC3 punctate autophagosome structures following activation of canonical autophagy by nutrient starvation (Fig 1A). To investigate non-canonical autophagy, we analysed LC3-associated phagocytosis (LAP), where LC3 is lipidated to zymosan-containing single-membrane phagosomes (Florey *et al*, 2011; Martinez *et al*, 2011; Romao *et al*, 2013). Interestingly, we detected ATG16L1 recruitment to LC3-positive phagosomes in the mouse macrophage cell line J7741.A (Fig 1B). To broaden this observation, we also analysed models of drug-induced non-canonical autophagy. We have previously reported activation of a non-canonical autophagy pathway by the sodium/proton ionophore monensin, which promotes LC3 lipidation to acidic single-membrane endolysosomal compartments, including those generated following entosis, a live cell engulfment process or engulfment of plain latex beads (Florey *et al*, 2015; Jacquin *et al*, 2017). Upon monensin treatment, we observed both ATG16L1 recruitment and LC3 lipidation to large entotic corpse-containing vacuoles (Fig 1C) and to latex bead-containing phagosomes (Fig 1D). These data demonstrate that, like double-membrane autophagosomes in canonical autophagy, ATG16L1 is recruited to single-membrane compartments during non-canonical autophagy.

ATG16L1 recruitment to autophagosomes is dependent on PI3P generated by VPS34, and the PI3P effector WIPI2b that directly

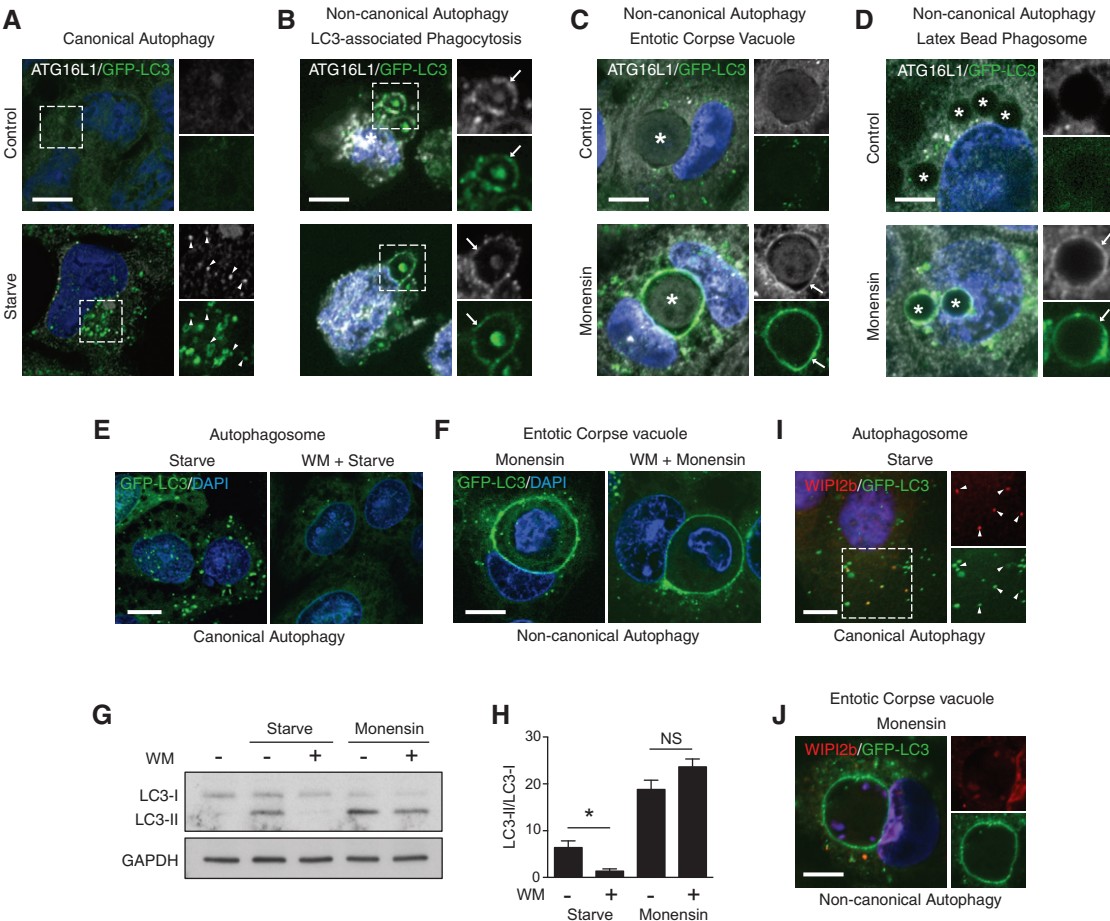

**Figure 1.   ATG16L1 is recruited to endolysosomal membranes during non-canonical autophagy.**

A     Confocal images of control and starved HCT116 cells expressing GFP-LC3 and stained for ATG16L1. Arrows indicate autophagosome puncta double positive for LC3 and ATG16L1. Scale bar: 10 μm.

B–D   Confocal images of ATG16L1 and GFP-LC3 on (B) zymosan-containing phagosomes in J774A.1 cells (arrows indicate phagosomes), scale bar: 5 μm; (C) monensin-treated entotic corpse vacuoles in MCF10A cells (asterisk indicate entotic corpse, arrows indicate entotic vacuoles), scale bar: 10 μm; and (D) latex bead-containing phagosomes in monensin-treated HCT116 cells (asterisk indicate bead-containing phagosomes, arrows indicate phagosome membranes), scale bar: 5 μm.

E, F   Confocal images of GFP-LC3 in (E) starved cells or (F) entotic corpse vacuoles in monensin-treated MCF10A cells ± wortmannin pretreatment. Scale bars: 10 μm.

G     Western blotting of LC3 in control, starved or monensin-treated HEK293 cells ± wortmannin.

H     Quantification of LC3-II/LC3-I ratios from (G).

I, J   Confocal images of WIPI2b staining and GFP-LC3 in (I) starved HCT116 cells. Arrows indicated double-positive autophagosome structures, and (J) entotic corpse vacuoles in monensin-treated MCF10A cells. Scale bars: 10 μm.

Data information: In (H), data are presented as mean + SEM from three separate experiments. *$P < 0.04$ (Student's *t*-test).

binds ATG16L1 (Dooley *et al*, 2014). In agreement with this, pretreatment of cells with the PI3 kinase inhibitor wortmannin abolishes canonical autophagy induced by starvation as measured by LC3 puncta formation and lipidation (Fig 1E and G). However, in line with our previous report (Florey *et al*, 2015), total levels of LC3 lipidation and localisation to entotic corpse vacuoles following treatment with monensin were not inhibited by wortmannin (Fig 1F–H). Consistent with the dispensability of VPS34 and PI3P in monensin-induced non-canonical autophagy, we found no evidence of WIPI2b recruitment to LC3-positive entotic corpse vacuoles (Fig 1J), while WIPI2b was observed at LC3-positive starvation-induced autophagosomes (Fig 1I).

Together these data show that, upon activation of non-canonical autophagy, ATG16L1 can be recruited to single-membrane endolysosomal compartments independently of PI3P and WIPI2b, and thus through a mechanism distinct from canonical autophagy.

### ATG16L1 structure function in canonical autophagy

To investigate the novel mechanisms underlying ATG16L1 recruitment to membranes during non-canonical autophagy, we sought to map the domain of ATG16L1 required to support endolysosomal LC3 lipidation. To do so, we re-expressed a set of ATG16L1 constructs (depicted in Fig 2A) in multiple independently generated ATG16L1-deficient cell lines. The ATG16L1 constructs consist of full-length ATG16L1 (FL), ATG16L1 lacking the region 219–242 that contains the WIPI2b and FIP200 binding sites (ΔFBD), and ATG16L1 lacking the WD40 CTD (ΔWD). We engineered ATG16L1-deficient

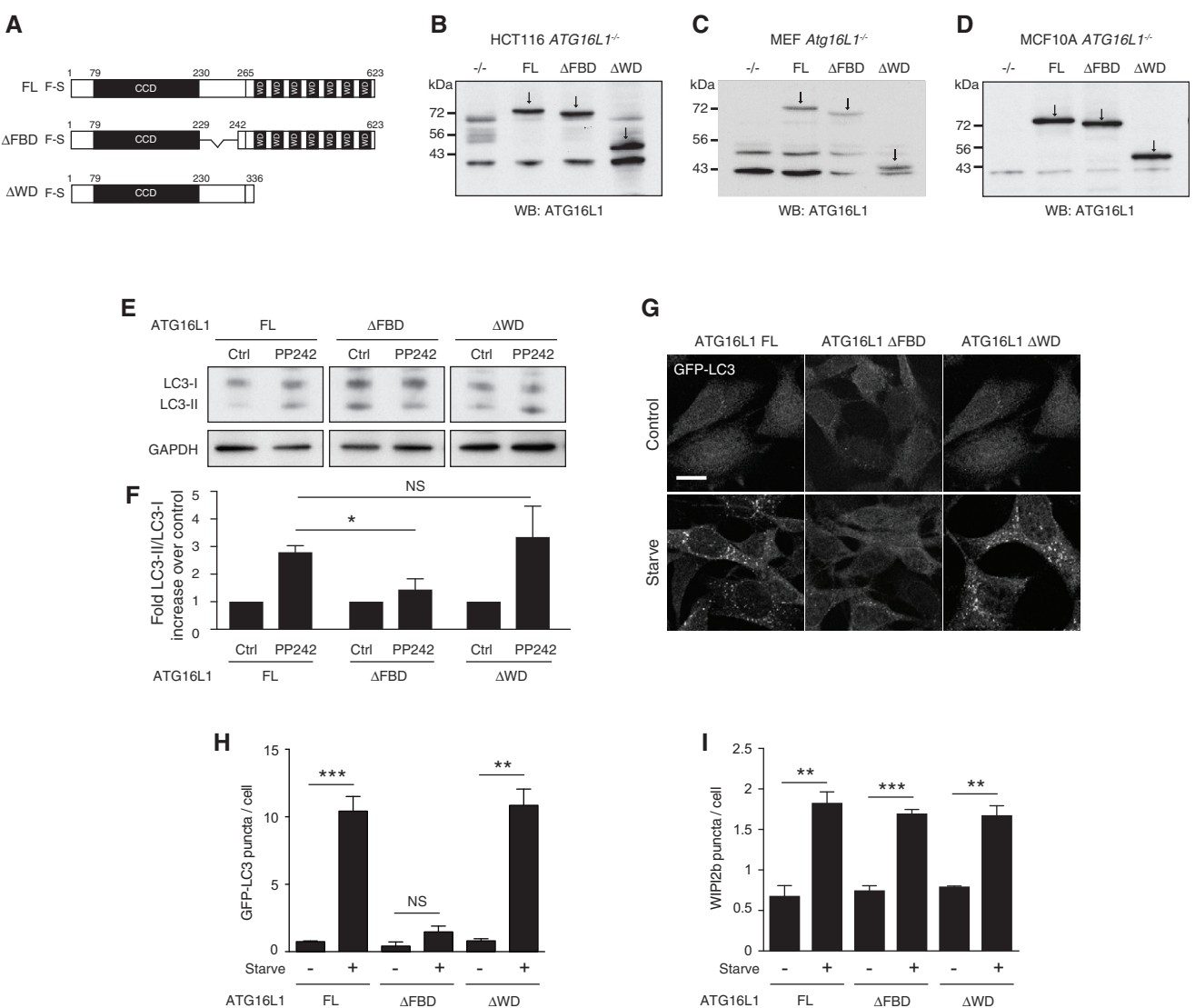

**Figure 2. The WD domain of ATG16L1 is not required for canonical autophagy.**

A   Diagram of full-length (FL) 229–242 deletion (ΔFBD) and 1–336 (ΔWD) ATG16L1 constructs used in this study.

B-D   Western blot analysis of ATG16L1 in (B) HCT116 *ATG16L1*$^{-/-}$, (C) MEF *Atg16L1*$^{-/-}$ and (D) MCF10A *ATG16L1*$^{-/-}$ cells stably re-expressing ATG16L1 constructs. Arrows indicate specific ATG16L1 band.

E   Western blotting for LC3 in complemented HCT116 cells ± PP242 (1 μM, 1 h).

F   Quantification of fold differences of LC3-II/LC3-I ratios over controls from (E).

G   Confocal images of GFP-LC3 in complemented MEF cells ± starvation (1 h). Scale bar: 10 μm.

H   Quantification of GFP-LC3 puncta from 100 MEF cells per experiment cultured in full media (control) or EBSS (starve) for 1 h.

I   Quantification of WIPI2b puncta in ATG16L1-complemented HCT116 cells. Puncta from 100 cells were counted per experiment.

Data information: Data represent mean ± SEM from three separate experiments. (F) *$P < 0.02$. (H) ***$P < 0.0001$, **$P < 0.001$. (I) ***$P < 0.0006$, **$P \ll 0.005$ (Student's *t*-test).

clones of the human colon cancer cell line HCT116 and the human breast epithelial cell line MCF10A using CRISPR/Cas9. We also utilised ATG16L1-deficient mouse embryonic fibroblasts (MEFs) previously generated by traditional methods based on homologous recombination. In all three cases, we were able to generate stable cell lines deficient in ATG16L1 and in which either full-length or truncated ATG16L1 could be expressed (Fig 2B-D).

As expected, cells expressing full-length ATG16L1 exhibited an increase in lipidated LC3 (LC3-II) following activation of canonical autophagy by mTOR inhibition, using PP242 (Fig 2E and F). ΔWD

cells displayed similar LC3 lipidation levels to full-length expressing cells (Fig 2E and F), indicating this domain is not required. In line with previous reports, however, we saw a reduction of LC3 lipidation in ΔFBD cells lacking the WIPI2b and FIP200 binding sites (Fig 2E and F), indicating this domain is required for canonical autophagy (Gammoh *et al*, 2013). Consistent with the Western blot results, we also observed increases in autophagosome number, as assessed by GFP-LC3 puncta, in full-length and ΔWD, but not ΔFBD MEFs (Fig 2G and H) and HCT116 cells (Fig EV1) following starvation. Formation of WIPI2b puncta lies upstream of ATG16L1

recruitment and thus LC3 lipidation in the canonical autophagy pathway. Consistent with this established hierarchy of autophagy proteins, all ATG16L1-expressing HCT116 cell lines supported increased WIPI2b puncta formation following starvation (Fig 2I).

These data demonstrate that our complemented ATG16L1 cell lines are competent for canonical autophagy induction and confirm previously reports data that the WIPI2b and FIP200 binding domain of ATG16L1 is required for LC3 lipidation and association with autophagosomal membranes in starvation-induced canonical autophagy, while the WD40 CTD is dispensable for this process.

### ATG16L1 structure function in monensin-induced non-canonical autophagy

We next used our set of complemented ATG16L1 cell lines to study LC3 lipidation induced during monensin driven non-canonical autophagy. We have previously shown that in wild-type cells, the ionophore monensin increases lipidated LC3 (LC3-II) via two parallel pathways. Firstly, by inserting into membranes and facilitating the exchange of sodium and hydrogen ions, monensin raises lysosome pH thus blocking autophagosome flux (canonical). At the same time, monensin also induces osmotic imbalances within endolysosomal compartments, which are then targeted for lipidation with LC3 (non-canonical autophagy pathway; Florey *et al*, 2015). In contrast, the V-ATPase inhibitor bafilomycin A1 increases LC3-II levels solely by inhibiting autophagosome flux. Thus, comparing the levels of lipidated LC3 (LC3-II) induced by monensin versus bafilomycin A1 allows the distinction between these parallel effects. Accordingly, in HCT116 cells expressing full-length ATG16L1 monensin induces significantly more LC3-II than bafilomycin A1, (Fig 3A), indicating the activation of non-canonical autophagy. A similar pattern is observed in ΔFBD cells (Fig 3A), suggesting the

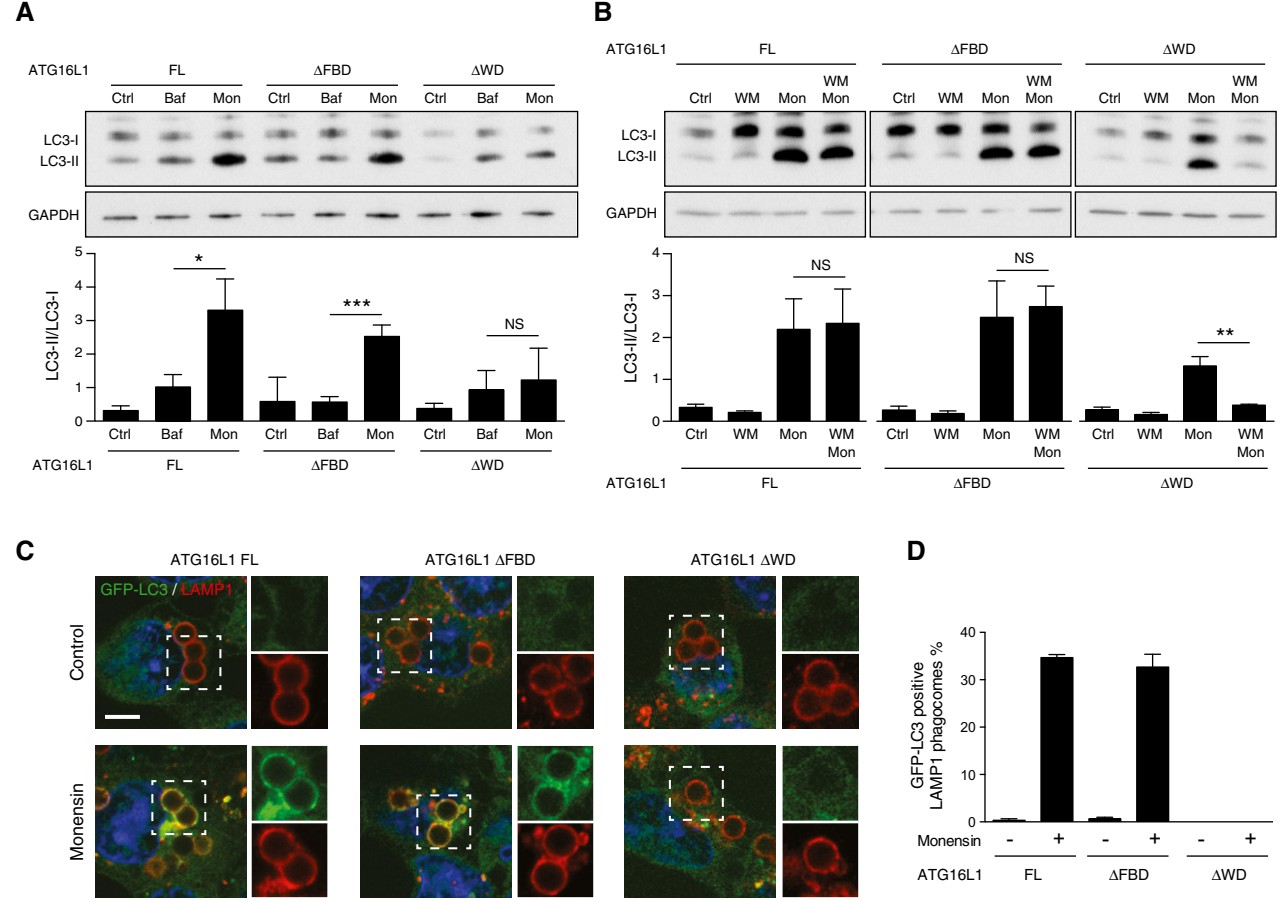

**Figure 3. The WD domain of ATG16L1 is required for monensin-induced non-canonical autophagy.**

A  Western blotting of LC3 in ATG16L1 FL, ΔFBD- or ΔWD-complemented HCT116 cells treated with bafilomycin (100 nM) or monensin (100 μM) for 1 h. Below is the quantification of LC3-II/LC3-I ratios.

B  Western blotting of LC3 from ATG16L1-complemented HCT116 cells treated with wortmannin (67 μM), monensin (100 μM) or both for 1 h. Below is the quantification of LC3-II/LC3-I ratios.

C  Confocal images of latex bead-containing phagosomes in control and monensin-treated GFP-LC3-expressing HCT116 cells complemented with ATG16L1 FL, ΔFBD or ΔWD. Samples were stained for LAMP1. Cropped images show bead phagosomes. Scale bar: 5 μm.

D  Quantification of GFP-LC3 recruitment to LAMP-1-positive phagosomes. 100 phagosomes were counted per experiment.

Data information: In (A, B, D), data are presented as mean ± SEM from three separate experiments. (A) *P < 0.02, ***P < 0.001. (B) **P < 0.0002 (Student's *t*-test).

FBD domain is dispensable in this context. Strikingly, we detected no difference in LC3-II levels between bafilomycin A1 and monensin treatment in ΔWD cells (Fig 3A). These data provide the first indication that, in contrast to canonical autophagy, the WD40 CTD of ATG16L1 is required for non-canonical autophagy.

LC3 lipidation associated with non-canonical autophagy induced by monensin is resistant to wortmannin in wild-type cells (Fig 1D–G). We observe similar wortmannin-resistant LC3 lipidation in full-length and ΔFBD cells following monensin treatment (Fig 3B). However, in ΔWD cells, wortmannin significantly inhibited monensin driven LC3 lipidation. These data suggest that the lipidated LC3 observed in ΔWD cells derives only from a wortmannin-sensitive canonical autophagy pathway.

In order to better differentiate between canonical versus non-canonical autophagy pathways, we next examined the localisation of LC3 in our set of complemented cells. Monensin mediated the recruitment of GFP-LC3 to LAMP1-positive latex bead-containing phagosomes in full-length and ΔFBD cells (Fig 3C and D). We have previously shown that the recruitment of GFP-LC3 in this model is lipidation dependent and not associated with canonical autophagy (Florey et al, 2015). In ΔWD cells, we could detect no GFP-LC3 recruitment to phagosomes (Fig 3C and D). A similar result was seen when examining entotic corpse vacuoles in complemented $ATG16L1^{-/-}$ MCF10A cells treated with monensin (Fig EV2A and B). This is consistent with monensin-induced LC3 lipidation being driven by continuous recruitment of the ATG16L1 complex to endolysosomal membranes rather than inhibition of ATG4 activity. This model is supported by fluorescence recovery after photobleaching (FRAP) data that shows GFP-LC3 localisation to monensin-treated entotic corpse vacuoles reappears following photobleaching (Fig EV2C and D). These experiments demonstrate that cells lacking the WD40 CTD of ATG16L1 are unable to support LC3 lipidation to endolysosomal compartments associated with monensin-induced non-canonical autophagy.

## ATG16L1 structure function in physiological non-canonical autophagy

We next sought to test the requirement of the WD40 CTD of ATG16L1 in more physiological examples of non-canonical autophagy. LC3-associated phagocytosis (LAP) occurs during the phagocytic engulfment of apoptotic and necrotic cells, or the engulfment of some fungal and bacterial pathogens. LC3 is targeted to these single-membrane phagosomes independently of the canonical autophagy pathway, but dependent on the lipidation machinery that includes ATG16L1. MEF cells are able to engulf apoptotic cells (Gardai et al, 2005) and have previously been shown to be competent for LAP (Hubber et al, 2017). Using live cell imaging, we detected GFP-LC3 recruitment to phagosomes containing CellTracker Red-labelled apoptotic corpses in full-length and ΔFBD MEFs. However, consistent with our previous data using monensin, ΔWD cells did not support GFP-LC3 recruitment to apoptotic corpse-containing phagosomes (Fig 4A and B). These data demonstrate an essential requirement for the WD40 CTD of ATG16L1 during LC3-associated phagocytosis.

Similar to LAP, LC3 has also been shown to be targeted to newly formed macropinosomes via a non-canonical autophagy pathway (Florey et al, 2011). Using red dextran as a fluid phase marker in

PDGF stimulated MEFs, we found GFP-LC3 recruitment to red-labelled macropinosomes in full-length and ΔFBD cells but not in ΔWD cells (Fig 4C). Further, non-canonical autophagy is induced by vacuolating toxin A (VacA); a virulence factor secreted by the pathogen Helicobacter pylori. VacA inserts into and oligomerises in target cell plasma membranes and is then internalised through endocytosis. Once internalised, VacA acts as a chloride channel creating ionic and electrochemical imbalances within the endocytic compartment. V-ATPase activity is upregulated to counter these changes, which in turn promotes accumulation of weak base amines in the endocytic compartment resulting in osmotic vacuolation, and subsequent recruitment of LC3 (Florey et al, 2015). As expected, stimulation with VacA and $NH_4Cl$ resulted in profound vacuolation in all complemented ATG16L1 MEF cell lines. However, only full-length and ΔFBD cells exhibited GFP-LC3-positive vacuoles, while vacuoles in ΔWD cells remained GFP-LC3 negative (Fig 4D). Together, these data confirm that the WD40 CTD of ATG16L1 is essential for LC3 lipidation in a range of engulfment processes which activate non-canonical autophagy.

## Recruitment of ATG16L1 to membranes during non-canonical autophagy is dependent on the WD40 CTD

We have taken distinct physiological processes that are known to activate non-canonical autophagy and demonstrated an essential requirement for the WD40 CTD of ATG16L1 in all cases. To rule out the possibility that this is due to a failure to form the E3-like ATG5-ATG12-ATG16 protein complex, we analysed full-length and ΔWD ATG16L1 immunoprecipitates using Western blotting and mass spectrometry. We found both constructs are competent in binding ATG5 and ATG12 (Fig 5A and B). Coupled with the fact that ΔWD ATG16L1 cells can support LC3 lipidation to autophagosome structures upon nutrient starvation, we conclude that the ATG16L1 ΔWD protein retains its structural integrity and another explanation must exist for its lack of function during non-canonical autophagy.

The role of ATG16L1 in canonical autophagy is to target the E3-like ATG12-5 16 complex to forming autophagosome membranes. Unlike the full-length protein, ATG16L1 ΔWD failed to recruit to latex bead phagosomes following monensin treatment (Fig 5C). In support of this, using fractionation and Western blotting, we detected increased amounts of ATG16L1 and ATG5 in membrane fractions from full-length expressing cells following monensin treatment but not in ΔWD-expressing cells (Fig 5D–G). Indeed, under resting conditions there appeared to be less ATG16L1 ΔWD in the membrane fraction compared to full-length ATG16L1. These data suggest that the WD40 CTD of ATG16L1 is required to target the protein to membrane compartments during non-canonical autophagy.

## Identification of key residues on the top face of ATG16L1 WD40 CTD required for non-canonical autophagy

To gain more molecular insight into how the WD40 CTD of ATG16L1 regulates non-canonical autophagy, we used a published algorithm to predict residues important for supporting protein interactions with the top faces of WD40 proteins (Wu et al, 2012) and identified 12 candidates. To screen these residues for a role in

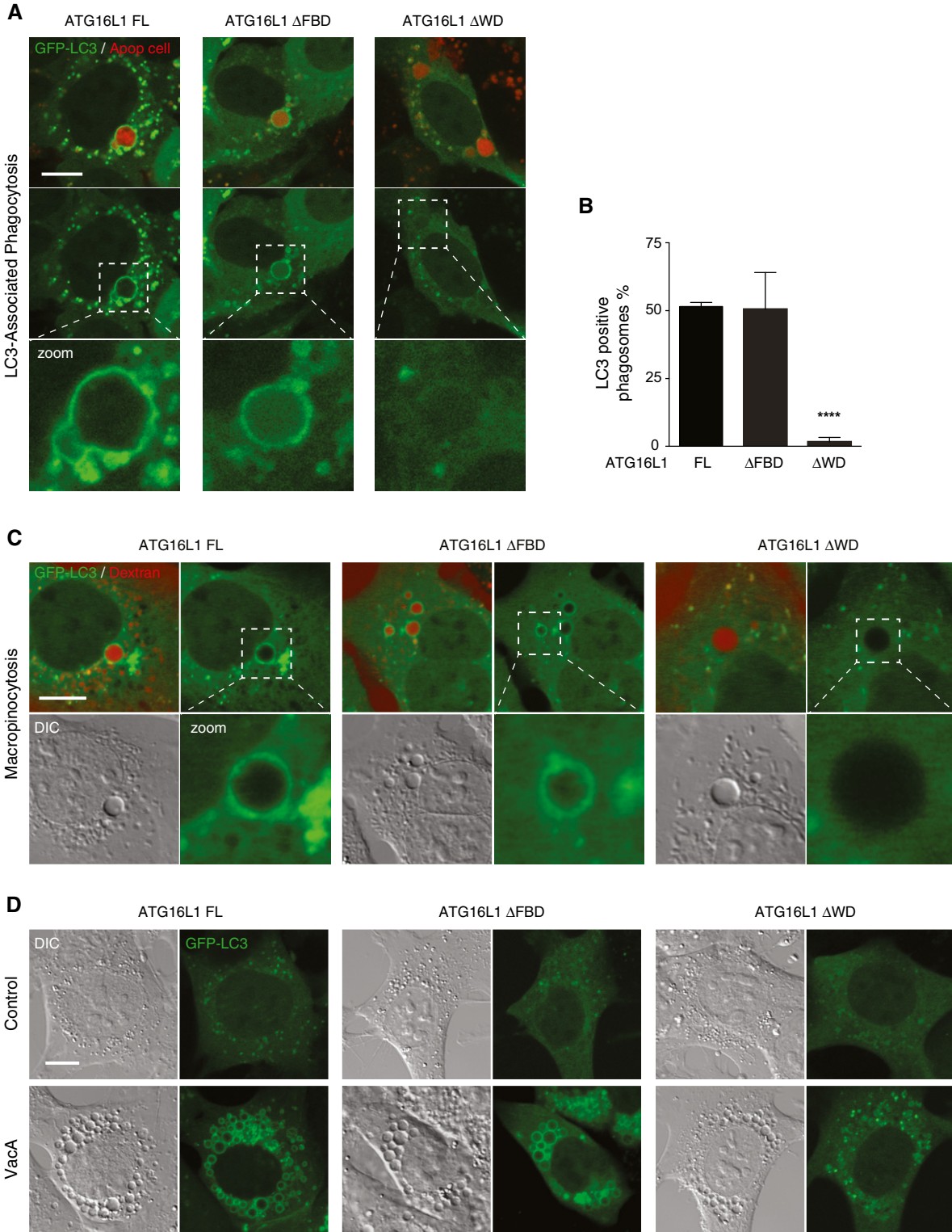

**Figure 4.  The WD domain of ATG16L1 is essential for LC3-associated phagocytosis and other non-canonical autophagy-dependent processes.**

A   Confocal images of GFP-LC3 in ATG16L1-complemented MEF cells phagocytosing red-labelled apoptotic cells. Scale bar: 10 μm.

B   Quantification of GFP-LC3 recruitment to apoptotic corpse-containing phagosomes in (A). Twenty phagosomes were counted per experiment.

C   Confocal images of red dextran-positive macropinosomes ATG16L1 complimented MEF cells. Cropped images show macropinosomes. Scale bar: 10 μm.

D   Confocal images of GFP-LC3 in ATG16L1-complimented MEF cells treated with VacA toxin (10 μM, 4 h). Scale bar: 10 μm.

Data information: In (B), data are presented as mean ± SEM from three separate experiments. ****P < 0.0001 (Student's *t*-test).

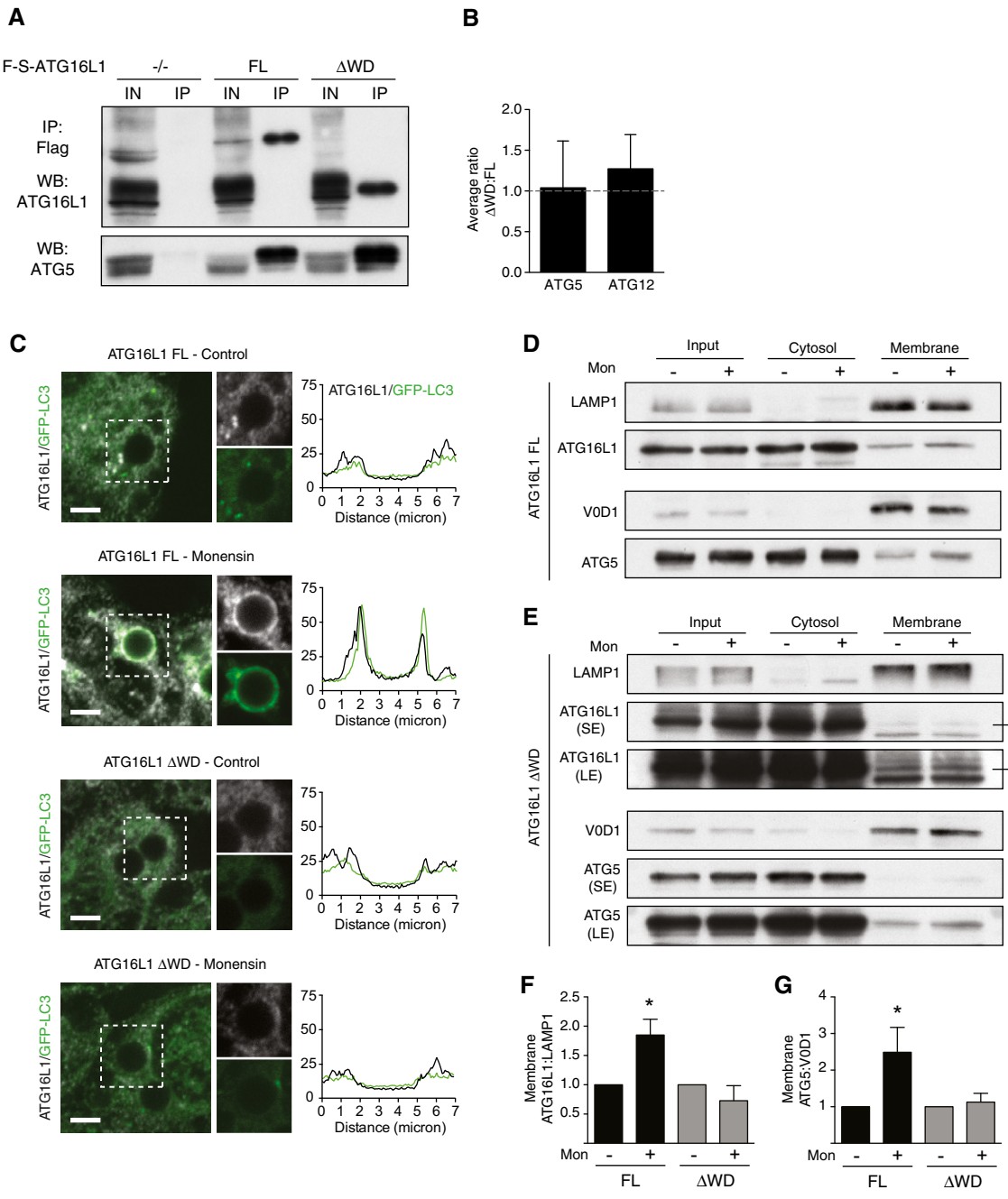

**Figure 5. The WD domain of ATG16L1 is required for its recruitment to single-membrane compartments during non-canonical autophagy.**

A    Analysis of ATG16L1 immunoprecipitation from *Atg16L1*$^{-/-}$, FL and ΔWD MEF cells treated with monensin. Input and IPs were probed for ATG16L1 and ATG5.

B    Mass spectrometry analysis of ATG5 and ATG12 protein levels pulled down with FL and ΔWD ATG16L1.

C    Confocal images of GFP-LC3 and ATG16L1 on latex bead-containing phagosomes in FL and ΔWD-expressing HCT116 cells ± monensin (100 μM, 1 h). Cropped images show phagosomes. Scale bar: 3 μm. Line profile analysis of ATG16L1 and GFP-LC3 fluorescence intensity is shown for representative phagosomes.

D, E    Western blot analysis of total lysate, cytosolic and membrane fractions from (D) FL and (E) ΔWD-expressing HCT116 cells ± monensin. Membranes were probed ATG16L1, ATG5 and membrane markers, LAMP1 and V0D1.

F, G    Quantification of membrane-associated (F) ATG16L1 and (G) ATG5 from experiments above, normalised to unstimulated conditions.

Data information: In (B, F, G), data are presented as mean ± SEM from on (B) three (F, G) separate experiments. *$P < 0.02$ (Student's *t*-test).

non-canonical autophagy, we generated *ATG16L1*$^{-/-}$ HCT116 cell lines stably re-expressing point mutants generated by site-directed mutagenesis. Using Western blotting, we then tested the ability of

wortmannin treatment to inhibit monensin-induced LC3 lipidation, similar to that used in Fig 3B. From our initial list, we found three residues N453, F467 and K490, which when mutated to alanine

displayed a robust inhibition of monensin-induced LC3 lipidation following wortmannin pretreatment (Fig EV3).

Using the recently derived crystal structure of the ATG16L1 WD40 CTD (Bajagic *et al*, 2017), we found that N453, F467 and K490 were in close proximity to one another and generated a pocket on the top face of the WD40 CTD (Fig 6A and B). These residues were also highly conserved through multiple species (Fig 6C). We chose two residues (F467 and K490) to study in more detail. Using HCT116 cells complemented with either full-length (FL) or F467A or K490A ATG16L1 (Fig 6D), we found they exhibited no defect in canonical autophagy induced by starvation as measured by GFP-LC3 puncta count or WIPI2b puncta formation (Fig 6E and F). However, MEF cells expressing F467A or K490A mutants showed a dramatic inhibition of LAP upon phagocytosis of apoptotic cells (Fig 6G and H), similar to that seen in ΔWD-expressing cells (Fig 4A). The F467A and K490A mutants were unable to recruit to latex bead-containing single-membrane phagosomes upon monensin treatment (Fig 6I and J), suggesting these sites are important in supporting ATG16L1 recruitment to these membranes. Interestingly, we found that the WD40 CTD alone could not recruit to phagosome membranes, which suggest the domain is necessary but not sufficient for recruitment of the complex (Fig 6K and L). Together, these data reveal for the first time specific sites within the WD40 CTD of ATG16L1, which are important for its role in non-canonical autophagy.

### Inhibition of non-canonical autophagy by deleting the WD40 CTD of ATG16L1 inhibits MHC class II antigen presentation in dendritic cells

While the full extent of non-canonical autophagy function in cells is not fully understood, it has been implicated in a number of immune-related processes including pathogen clearance (Sanjuan *et al*, 2007; Ma *et al*, 2012) and presentation of exogenous antigens (Lee *et al*, 2010; Ma *et al*, 2012). Commonly this has been achieved by inhibiting the LC3 conjugation machinery, that is ATG5 or ATG7. We now sought to more specifically test the role of non-canonical autophagy in antigen presentation using the ATG16L1 ΔWD system. To achieve this, we utilised a recently engineered mouse model where ATG16L1 is truncated at position E230 and thus lacks the WD40 CTD. This truncated version of ATG16L1 is expressed in all cells (Fig 7A), and unlike *Atg16L1* knockout mice, E230 mice are viable, which suggest they remain competent for canonical autophagy. To test for non-canonical autophagy, we examined zymosan phagocytosis in bone marrow-derived dendritic cells (BMDCs). We found LC3 recruitment to zymosan-containing phagosomes in wild-type BMDCs but found no LC3 recruitment to phagosomes in E230 cells (Fig 7B). This demonstrates E230 BMDCs are deficient in LC3-associated phagocytosis. We next examined the presentation of an exogenous antigen. BMDCs were incubated with GFP-Eα peptide for 24 h and analysed for Eα peptide presentation on MHC class II molecules by flow cytometry using an antibody that recognises the peptide in complex with MHCII (Macritchie *et al*, 2012). We found that E230 BMDCs displayed a significant inhibition in presentation of exogenous antigen as compared to wild-type BMDCs (Fig 7C). This defect was not due to impaired uptake of antigen, as both wild-type and E230 BMDCs showed a dose-dependent increase in GFP signal following incubation with GFP-Eα peptide (Fig 7D). Indeed,

E230 BMDCs displayed increased GFP signal as compared to wild-type cells, which potentially points to a defect in antigen processing as GFP signal can be lost as the peptide is processed. These data demonstrate a functional consequence of inhibiting non-canonical autophagy through the targeting of the ATG16L1 WD40 CTD. It verifies a role for non-canonical autophagy in antigen presentation and provides a clean system with which to study this process.

### Influenza A infection activates non-canonical autophagy via a proton channel

Having established the essential role of the WD40 CTD of ATG16L1 in the activation of endolysosomal LC3 lipidation, and as a method to distinguish between canonical autophagy and non-canonical autophagy pathways, we utilised our cell lines to demonstrate a novel non-canonical autophagy-dependent process. Considering our previous data showing activation of non-canonical autophagy through alterations in ion or proton gradients, either by ionophore drugs or pathogenic factors such as VacA, we tested the effect of another physiological modulator of ion movement. During influenza A virus infection, the viral protein M2 inserts into cell membranes of infected cells and acts as a highly selective proton channel. Expression of M2 has previously been shown to induce LC3 lipidation at both the plasma membrane of infected cells and in perinuclear structures (Beale *et al*, 2014). There are conflicting reports as to whether this depends on the proton channel activity of M2 (Gannage *et al*, 2009; Ren *et al*, 2015). The antiviral drug amantadine is a selective blocker of the M2 proton channel in sensitive strains. The laboratory-adapted influenza A virus (IAV) PR8 strain is resistant to amantadine, whereas IAV strain Udorn is sensitive. We therefore tested the ability of PR8 or a derivative bearing segment 7 (and hence the sensitive M2) from Udorn (MUd) to relocalise LC3 in the presence or absence of amantadine. We found that the IAV strain, which is sensitive to amantadine completely fails to relocalise LC3 (Fig 8A). Our results confirm that the proton channel activity of M2 is required for LC3 relocalisation. Next, we tested the requirement for ATG16L1 FBD and WD40 domains. We determined that full-length or ΔFBD ATG16L1 were able to complement ATG16L1 deficiency, but ΔWD40 or K490A ATG16L1 were unable to do so (Fig 8B). We further showed a dependency on the WD40 CTD of ATG16L1 for IAV-induced LC3 lipidation by Western blotting (Fig 8C). These data indicate that IAV-induced LC3 lipidation is driven predominantly by non-canonical autophagy. We found no effect on viral titres in the absence of non-canonical autophagy (Fig EV4), which suggests further work is required to elucidate the function of this pathway during influenza infection.

## Discussion

In this study, we identified a novel mechanism that recruits ATG16L1 to sites of LC3 lipidation during non-canonical autophagy processes, including LC3-associated phagocytosis. The C-terminal WD domain of ATG16L1 is essential in its targeting, in complex with ATG5-ATG12, to single-membrane compartments rather than double-membrane autophagosomes. We also identified, for the first time, specific sites within the top face of the WD40 CTD required for the non-canonical activity of ATG16L1. Our results support

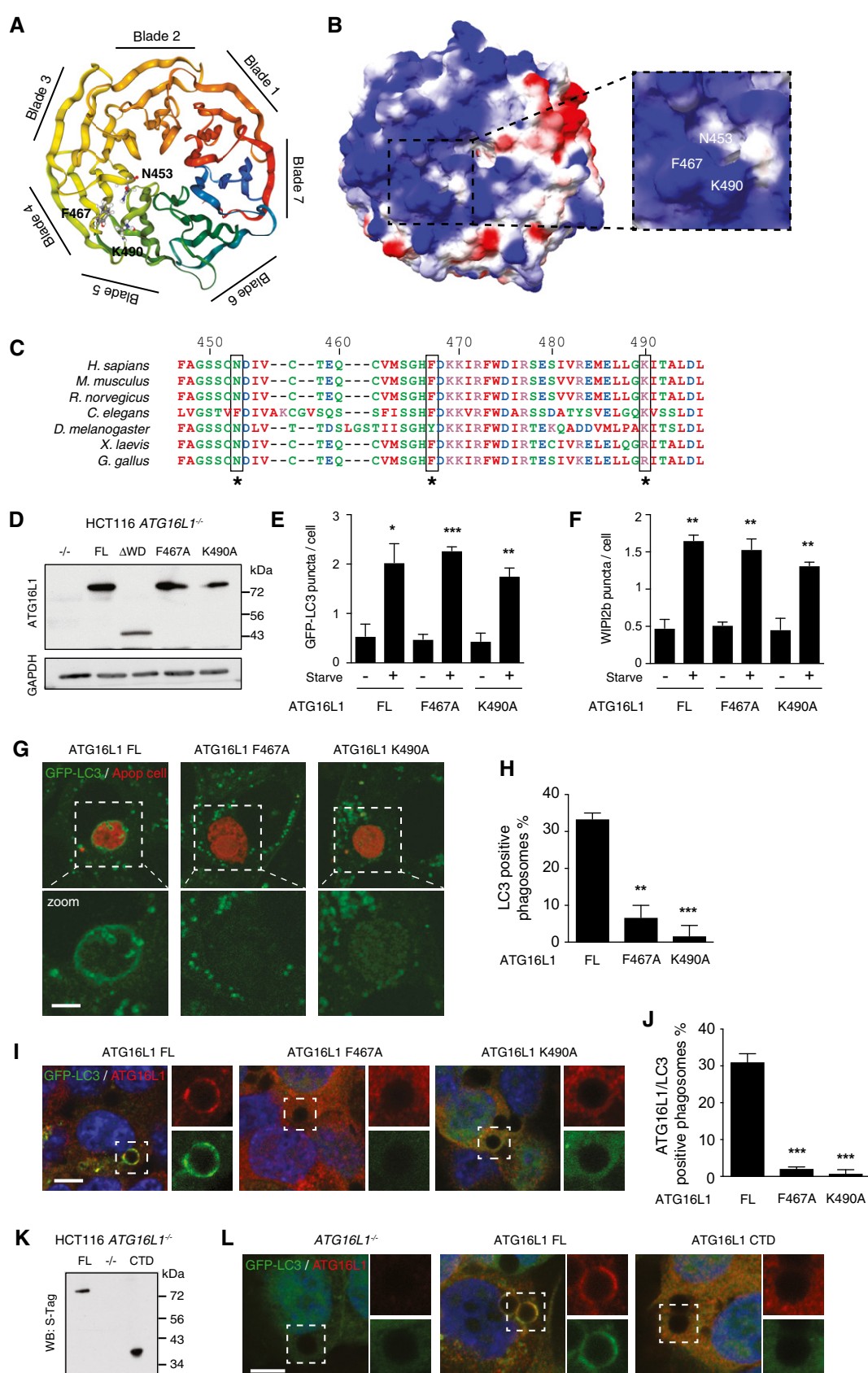

Figure 6.

**Figure 6.  ATG16L1 recruitment and function in non-canonical autophagy requires specific residues within the WD40 CTD.**

A    Ribbon model of the top face of ATG16L1 WD40 CTD with critical residues in ball and stick. Structural image generated in NGL viewer using Protein Database (PDB) 5NUV.

B    Surface of ATG16L1 WD40 CTD coloured to electrostatic potential (blue positive 2, red negative −2). Cropped image a zoom of the critical residues. Image generated in Swiss-PdbViewer.

C    Annotated alignment of ATG16L1 sequences (447–496) over different species.

D    Western blot analysis of ATG16L1 in HCT116 *ATG16L1*$^{-/-}$ cells stably re-expressing ATG16L1 constructs.

E    Quantification of GFP-LC3 puncta from 100 HCT116 cells per experiment cultured in full media (control) or EBSS (starve) for 1 h.

F    Quantification of WIPI2b puncta from 100 HCT116 cells per experiment cultured in full media (control) or EBSS (starve) for 1 h.

G    Confocal images of GFP-LC3 in ATG16L1-complemented MEF cells phagocytosing red-labelled apoptotic cells. Scale bar: 5 μm.

H    Quantification of GFP-LC3 recruitment to apoptotic corpse-containing phagosomes in (G). Twenty phagosomes were counted per experiment.

I    Confocal images of GFP-LC3 and ATG16L1 on latex bead-containing phagosomes in FL, F467A- and K490A-expressing HCT116 cells ± monensin (100 μM, 1 h). Cropped images show phagosomes. Scale bar: 5 μm.

J    Quantification of ATG16L1/GFP-LC3-positive phagosomes from (I).

K    Western blot analysis of ATG16L1 in HCT116 *ATG16L1*$^{-/-}$ cells stably re-expressing full-length (FL) and CTD (336-623) ATG16L1 constructs.

L    Confocal images of GFP-LC3 and ATG16L1 stained with anti-S-Tag antibodies on latex bead-containing phagosomes in knockout, FL and CTD expressing HCT116 cells ± monensin (100 μM, 1 h). Cropped images show phagosomes. Scale bar: 5 μm

Data information: In (E, F, H, J), data are presented as mean ± SEM from three separate experiments. (E) *P < 0.03, **P < 0.005, ***P < 0.0001. (F) **P < 0.005. (H) **P < 0.002, ***P < 0.0002. (J) ***P < 0.0003 (Student's *t*-test).

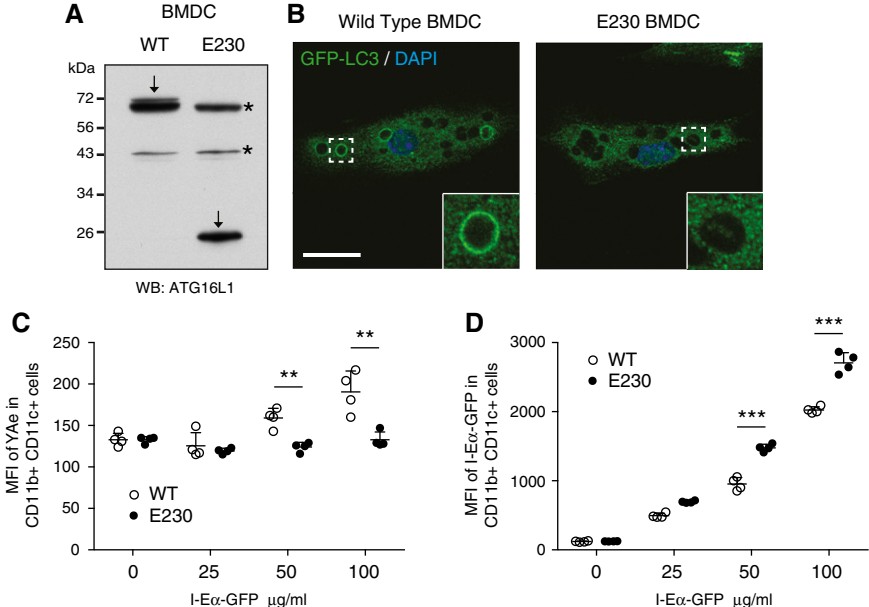

**Figure 7.  ATG16L1 WD40 CTD supports presentation of exogenous antigen by dendritic cells.**

A    Western blot analysis of ATG16L1 in wild-type and E230 BMDCs. Arrows indicate ATG16L1, and asterisks mark non-specific band.

B    Confocal images of LC3 in BMDCs phagocytosing zymosan particles. Insert shows phagosome. Scale bar: 20 μm.

C    Mean fluorescent intensity (MFI) FACs analysis of Y-Ae in wild-type (open circles) and E230 (filled circles) BMDCs exposed to different concentrations of Eα-GFP.

D    Mean fluorescent intensity (MFI) FACs analysis of Eα-GFP in wild-type (open circles) and E230 (filled circles) BMDCs exposed to different concentrations of Eα-GFP.

Data information: In (C, D), data are presented as mean ± SD from four replicates. Data are representative of three independent experiments. (C) **P < 0.002. (D) ***P < 0.0001 (Student's *t*-test).

published data that the WD40 CTD of ATG16L1 is dispensable for canonical autophagy induction. Indeed, yeast Atg16 lacks the WD40 CTD, consistent with the finding that this domain is not required for canonical autophagy, and suggesting that non-canonical autophagy may have evolved in higher organisms. Conversely, we show that the FBD domain required for canonical autophagy is dispensable for non-canonical autophagy.

The existence of a distinct mechanism for ATG16L1 recruitment during non-canonical autophagy is further supported by experiments that show that, unlike canonical autophagy, PI3P generation and VPS34 activity are not required for non-canonical autophagy-associated LC3 lipidation. At first, this may appear to contradict recent work showing LAP is dependent on Rubicon-mediated VPS34 activity and PI3P (Martinez *et al*, 2015). However, it is possible that in the context of phagocytosis, VPS34 and PI3P are required at an upstream step to mature the phagosome to a state competent for LC3 lipidation, without being directly involved in ATG16L1 recruitment. Phagocytosis is a complex process, and no clear distinction

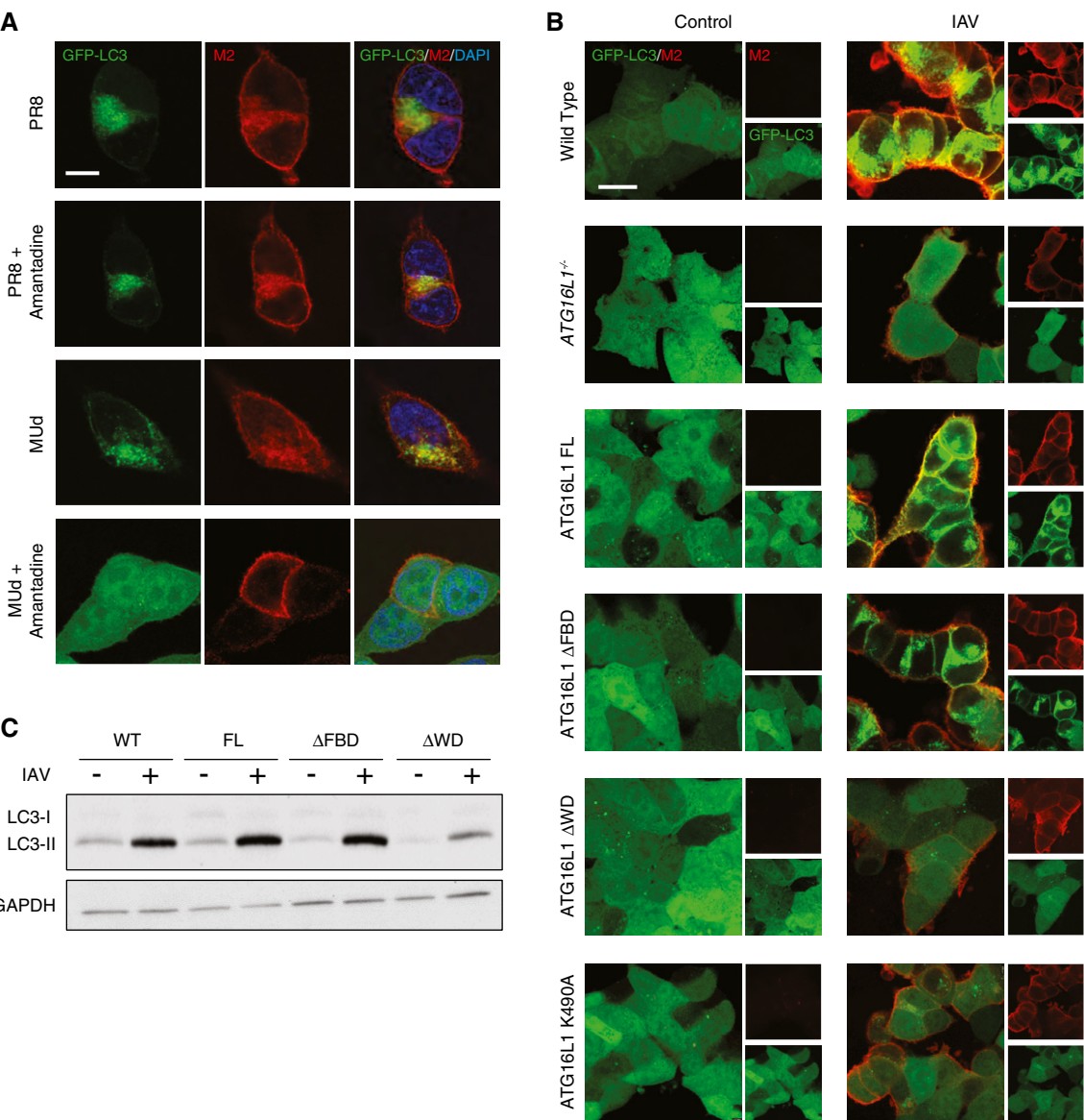

**Figure 8. Influenza infection activates non-canonical autophagy via proton channel activity of M2.**

A   Confocal images of GFP-LC3-transduced HCT116 cells infected at MOI 1 with IAV strains PR8 (amantadine resistant) or MUd (amantadine sensitive). Amantadine was added 3 h postinfection. Samples were fixed at 16 h postinfection and stained for M2 (red) and with DAPI. Scale bar: 10 μm.

B   Confocal images of GFP-LC3-transduced wild-type or *ATG16L1*$^{-/-}$ HCT116 cells complemented with the indicated constructs. Samples fixed at 16 h postinfection with IAV PR8 at MOI of 5 and stained for M2 (red). Scale bar: 20 μm.

C   Western blot analysis of LC3 in influenza-infected *ATG16L1*$^{-/-}$ HCT116 cells complemented with the indicated constructs.

can be made between upstream maturation and direct autophagy protein recruitment. However, activating non-canonical autophagy using the ionophore monensin bypasses the VPS34-dependent maturation step as it targets already mature endolysosomal compartments. Thus, monensin allows us to interrogate more specifically the role of VPS34 and PI3P in ATG16L1 recruitment during non-canonical autophagy.

Our work provides a clear genetic means to distinguish between canonical and non-canonical autophagy pathways. Previously, cells deficient in only canonical autophagy (e.g. ULK1/2, FIP200, ATG13 knockout cells) have been compared to cells deficient in both

canonical and non-canonical autophagy (ATG5, ATG7, ATG16L1 knockout cells) to infer a role for the non-canonical autophagy (Kim *et al*, 2013; Martinez *et al*, 2016). More recently, in the context of phagocytosis, Rubicon was shown to differentially effect canonical autophagy and LAP (Martinez *et al*, 2015). However, loss of Rubicon may have particular effects on endosomal maturation, possibly indicating that the requirement for Rubicon is specific to LAP rather than the non-canonical autophagy pathway in general. We have now uncovered a system where truncations or point mutations in ATG16L1, a *bona fide* autophagy protein, render cells deficient in non-canonical autophagy while remaining competent for canonical

autophagy. This is true for all types of non-canonical autophagy processes tested. We propose that ATG16L1 acts as a regulatory hub to direct LC3 lipidation during all autophagy-related processes, with FIP200 and WIPI2b proteins driving canonical autophagy, and the WD40 CTD coordinating non-canonical autophagy.

The discovery that the ATG16L1 WD40 CTD supports non-canonical autophagy is consistent with the fact that WD repeat structures are known to be important for protein–protein interactions. Whether there is one universal mechanism to recruit ATG16L1 through the WD domain or whether there are process-specific mechanisms remains to be determined. It would appear, however, that the sites identified here are important in all tested examples of non-canonical autophagy. Interestingly, TMEM59 and TMEM166/EVA1 have been implicated in autophagy activation through the interaction with the WD40 CTD of ATG16L1 (Boada-Romero *et al*, 2013; Hu *et al*, 2016). However, as yet, it is not clear whether these represent true non-canonical autophagy pathways. A Crohn's disease-associated point mutation T300A is located near the C-terminal domain of ATG16L1 and has been implicated in affecting autophagic processes, including one dependent on the WD40 CTD (Kuballa *et al*, 2008; Lassen *et al*, 2014; Boada-Romero *et al*, 2016). However, we found no obvious difference in GFP-LC3 recruitment to entotic vacuoles in *ATG16L1*$^{-/-}$ HCT116 cells expressing T300 or T300A versions of ATG16L1 (Fig EV5A). Similarly, monensin was able to drive GFP-LC3 recruitment to latex bead-containing phagosomes equally well in both T300- and T300A-expressing cells (Fig EV5B and C). These results support previously reported data where the T300A mutation was shown not to affect LAP (Martinez *et al*, 2015). It is still possible that under some contexts, such as inflammatory bowel disease, an increased ATG16L1 cleavage influenced by the T300A polymorphism may affect ATG16L1 WD40 CTD-dependent non-canonical autophagy.

The extent to which non-canonical autophagy is important within biological systems is currently unclear. Our data provide a clear strategy to identify and explore the consequences of processes that activate and depend on non-canonical autophagy. The ATG16L1 E230 mouse model provides a new *in vivo* system to investigate the physiological and pathophysiological roles of non-canonical autophagy. Indeed, we have been able to confirm a role for LAP in dendritic cell presentation of exogenous antigens on MHC class II. Previous studies have demonstrated that influenza A virus-induced LC3 lipidation was independent of FIP200 levels and could be detected at the plasma membrane of infected cells (Beale *et al*, 2014). These data would be consistent with non-canonical autophagy, but this possibility had not been explored. In this study, using our ATG16L1 mutants, we have now revealed M2 protein-dependent activation of non-canonical autophagy during influenza infection. This illustrates the possibility of distinguishing genetically between the canonical and non-canonical autophagy pathways by manipulating the WD40 CTD of ATG16L1.

Our results, and those of others, provoke a re-evaluation of data that has been interpreted assuming LC3 lipidation to be synonymous with canonical autophagy. This is particularly important in the context of host:pathogen interactions where invading microbes utilise ion channels to subvert host cell physiology, and where the host may employ LAP-like processes to effect cell-autonomous defence. We suggest that examining the dependence of LC3 lipidation on the ATG16L1 WD40 CTD in these contexts will provide a relatively simple means to distinguish whether the processes involved represent canonical or non-canonical autophagy. Further, by specifically inhibiting non-canonical autophagy, we can begin to tackle the important questions regarding the function and mechanisms of LC3 lipidation in this pathway.

# Materials and Methods

### Cell culture

HCT116 cells were cultured in McCoy's 5A (Lonza) with 10% foetal bovine serum (FBS Sigma), 1% penicillin and streptomycin. HCT116 T300A cells were kindly provided by Dr. David Boone (Indiana University School of Medicine). J774.A1, HEK293, MDCK (a kind gift from Dr. P. Digard) and mouse embryonic fibroblast (MEF) cells were cultured in Dulbecco's modified Eagle's medium (DMEM; Gibco Life Technologies) containing 10% foetal bovine serum (FBS Sigma), 1% penicillin and streptomycin. *Atg16L1*$^{-/-}$ MEFs were provided by Dr. Shizuo Akira (Osaka University). MCF10A cells were cultured in DMEM/F12 (Gibco) containing 5% horse serum (Sigma), EGF (20 ng/ml), hydrocortisone (0.5 mg/ml), cholera toxin (100 ng/ml) and insulin (10 μg/ml). All cells were maintained in an incubator at 37°C with 5% CO$_2$.

### Antibodies and reagents

Antibodies used in this study are anti-LC3A/B (4108, Cell Signalling, WB 1:1,000, IF 1:100), anti-ATG16L1 (PM040, MBL, IF 1:1,000; 8089, Cell Signalling, WB 1:1,000), anti-ATG5 (2630, Cell Signalling, WB 1:1,000), anti-GAPDH (25778, Santa Cruz, WB 1:2,000), anti-huLAMP1 (611043, Becton Dickinson, IF 1:100), anti-ATP6VV0d1 (56441, Abcam, WB 1:1,000), anti-WIPI2 (MCA5780GA, Bio-Rad, IF 1:100), anti-FlagM2 (F1804, Sigma), anti-M2 (ab5416, Abcam, IF 1:100), HRP-conjugated anti-mouse (7076, Cell Signalling, WB 1:1,000) or anti-rabbit (7074, Cell Signalling, WB 1:1,000). Alexa Fluor 568 anti-rabbit (A11011, Thermofisher, IF 1:500). Monensin (M5273), PP242 (P0037), wortmannin (W1628) and amantadine (A1260) were obtained from Sigma. Bafilomycin A1 (1334) was obtained from R&D Systems. PDGF was obtained from Peprotech (100-14B).

### IAV reverse genetics and infection

Stocks of influenza A virus PR8 (strain A/Puerto Rico/8/1934) and MUd, a resistant PR8 variant carrying segment 7 of IAV strain A/Udorn/307/1972 (Noton *et al*, 2007) were generated using the eight plasmid-based systems as previously described (de Wit *et al*, 2004) and propagated on MDCK cells. For infection, cells were first washed with serum-free DMEM and incubated with virus in serum-free DMEM at 37°C. After 1 h, the medium was replaced with DMEM containing 10% FBS. For immunofluorescence, imaging cells were fixed at 16 h p.i. using 4% paraformaldehyde in PBS.

### Plasmid construction and retroviral transduction

Flag-S-tagged Atg16L1 constructs were generated as previously described (Gammoh *et al*, 2013). Briefly, human ATG16L1 was inserted into pBabe Flag-S retroviral vector using SalI cloning sites.

Alanine point mutants were generated using QuikChange Site-directed Mutagenesis Kit (Stratagene). Primers see Table 1. Stable cell lines expressing ATG16L1 constructs were generated by retroviral transduction and selection as described previously (Gammoh et al, 2013). Briefly, cells were seeded and infected by centrifugation and stable cells were selected with puromycin (HCT116 0.8 μg/ml, MEF 1.5 μg/ml, MCF10A 2.5 μg/ml) for 2–5 days. HCT116 cells were transduced with a retroviral vector generated using M4P-GFP-LC3B (kind gift from F. Randow, Cambridge) and selected for eGFP expression by FACS-assisted cell sorting (Cell Phenotyping Hub, Department of Medicine, University of Cambridge).

**CRISPR/Cas9-mediated knockout of ATG16L1**

To generate ATG16L1 KO in MCF10A GFP-LC3 cells, gRNA sequence (GTGGATACTCATCCTGGTTC) with overhangs for containing a BpiI site was annealed and cloned into the pSpCas9(BB)-2A-GFP (Addgene, 48138; deposited by Dr. Feng Zhang) digested with the

**Table 1. Primers for site-directed mutagenesis of human ATG16L1 (623 amino acid isoform).**

| Mutant | Forward |
|---|---|
| E324A For | CGCATGACGGAGCGGTCAACGCAGTG |
| E324A Rev | CACTGCGTTGACCGCTCCGTCATGCG |
| N326A For | CATGACGGAGAGGTCGCCGCAGTGCAGTTCAG |
| N326A Rev | CTGAACTGCACTGCGGCGACCTCTCCGTCATG |
| M342A For | GCCACTGGAGGCGCGGACCGCAGGGTG |
| M342A Rev | CACCCTGCGGTCCGCGCCTCCAGTGGC |
| N386A For | CTTACCTATTAGCAGCTTCAGCTGATTTTGCAAGCCGAATC |
| N386A Rev | GATTCGGCTTGCAAAATCAGCTGAAGCTGCTAATAGGTAAG |
| K410A For | GGCCACAGCGGGGCAGTCCTCTCTGCC |
| K410A Rev | GGCAGAGAGGACTGCCCCGCTGTGGCC |
| L412A For | CACAGCGGGAAAGTCGCCTCTGCCAAGTTCC |
| L412A Rev | GGAACTTGGCAGAGGCGACTTTCCCGCTGTG |
| H428A For | GATTGTCTCAGGAAGTGCCGACCGGACCCTCAAAC |
| H428A Rev | GTTTGAGGGTCCGGTCGGCACTTCCTGAGACAATC |
| N453A For | GCAGGATCCAGCTGCGCTGACATTGTTTGCAC |
| N453A Rev | GTGCAAACAATGTCAGCGCAGCTGGATCCTGC |
| F467A For | GTGTAATGAGTGGACATGCTGACAAGAAAATTCGTTTCTG |
| F467A Rev | CAGAAACGAATTTTCTTGTCAGCATGTCCACTCATTACAC |
| K490A For | GATGAACTGTTAGGGGCGATCACTGCTCTGGAC |
| K490A Rev | GTCCAGAGCAGTGATCGCCCCTAACAGTTCATC |
| D536A For | CAAATGCGGCTCTGCCTGGACCCGGGTTG |
| D536A Rev | CAACCCGGGTCCAGGCAGAGCCGCATTTG |
| N581A For | CAGCTCTTCTATCGCTGCGGTGGCGTGGG |
| N581A Rev | CCCACGCCACCGCAGCGATAGAAGAGCTG |

BpiI restriction enzyme (Thermo Scientific, ER1011). The recombinant plasmid was introduced into MCF10A GFP-LC3 cells via AMAXA nucleofection (Kit V) along with a pBABE-puro construct (Addgene, 1764; deposited by Dr. Hartmut Land). Cells were selected with 2.5 μg/ml puromycin (P8833, Sigma) for 48 h, and single cell clones were obtained by limiting dilution. After clonal expansion, ATG16L1$^{-/-}$ clones were selected based on the absence of ATG16L1 protein as detected by Western blot.

To generate ATG16L1 KO in HCT116 cells, gRNA (ATTCTCTGCATTAAGCCGAT) was designed to target exons shared by all predicted transcripts with a high predicted activity (Doench et al, 2014) and specificity score (Hsu et al, 2014) and cloned into the BpiI site of pSpCas9(BB)-2A-puro V2.0 (Addgene, 62988; deposited by Dr Feng Zhang), cells were transfected using Lipofectamine 2000™ (Invitrogen) according to manufacturer's instructions and selected with puromycin (4 μg/ml), and single cell clones generated. Successful knockout of ATG16L1 was confirmed by Western blot and genome sequencing of the target site.

**Western blotting**

Cells were scraped into ice-cold RIPA buffer (150 mM NaCl, 50 mM Tris–HCl, pH 7.4, 1 mM EDTA, 1% Triton X-100 (Sigma, T8787), 0.1% SDS (Sigma, L3771), 0.5% sodium deoxycholate (Sigma, D6750) and lysed on ice for 10 min. Lysates were centrifuged for 10 min at 10,000 g at 4°C. Supernatants were then separated on 15 or 10% polyacrylamide SDS–PAGE gels and transferred to polyvinylidene difluoride membranes. Membranes were blocked in TBS-T supplemented with 5% BSA for 1 h at room temperature and incubated overnight at 4°C with primary antibodies diluted in blocking buffer. They were then incubated with a horseradish peroxidase-conjugated secondary antibody (Cell Signaling Technology, 7074S), and proteins were detected using enhanced chemiluminescence (GE Healthcare Life Sciences, RPN2209). Densitometry analysis was performed using ImageJ software.

**Immunoprecipitation**

MEF Atg16L1$^{-/-}$ cells expressing Flag-S-tagged ATG16L1 constructs were seeded into each of 4 × 144 mm tissue culture dishes per condition. After 48 h, cells were treated ± monensin with a final concentration of 100 μM for 1 h. Dishes were washed twice with ice-cold PBS and lysed in 2 ml of lysis buffer (50 mM Tris–HCl, 15 mM NaCl, 1 mM EDTA, 1% Triton X-100, 1 mM PMSF, 10 mM NaVO$_4$ and protease inhibitor cocktail). Samples were centrifuged at 4°C, 1,500 g for 15 min. The Triton-soluble material was pre-cleared for 1 h at 4°C using immunoglobulin G (IgG) conjugated 4% agarose bead slurry (Sigma). Pre-cleared supernatant was added to an S-protein agarose bead slurry (Novagen) for 6 h at 4°C. Samples were washed 3 times in lysis buffer and proteins eluted by addition of sample buffer followed by boiling for 5 min and analysed by Western blotting.

For some experiments, immunoprecipitated material was analysed by mass spectrometry using mass spectrometry. Briefly, samples were run a short distance (∼5 mm) into an SDS–PAGE gel, which was then stained with colloidal Coomassie stain (Imperial Blue, Invitrogen). The entire stained gel pieces were excised, destained, reduced, carbamidomethylated and digested overnight

with trypsin (Promega sequencing grade, 10 ng/μl in 25 mM ammonium bicarbonate). The resulting tryptic digests were analysed using LC-MS/MS on a system comprising a nanoLC (Proxeon) coupled to a LTQ Orbitrap Velos Pro mass spectrometer (Thermo Scientific). LC separation was achieved on a reversed-phase column (Reprosil C18AQ, 0.075 × 150 mm, 3 μm particle size), with an acetonitrile gradient (0–35% over 180 min, containing 0.1% formic acid, at a flow rate of 300 nl/min). Mass spectrometric data were processed using Proteome Discoverer v1.4 (Thermo Scientific) and searched against the mouse entries in Uniprot 2013.09.

## Membrane fractionation

HCT116 cells were seeded per condition on a 15-cm dish and cultured for 48 h. Cells put in suspension and treated with 100 μM monensin for 1 h. Input, cytosol and membrane fractions were isolated using the Mem-Per Plus Membrane Protein Extraction Kit (89842, Thermofisher) following product guidelines. Protein concentration was measured by BCA assay and equal amounts loaded onto polyacrylamide gels for SDS–PAGE analysis.

## Immunofluorescence

Unless otherwise indicated, immunofluorescence was performed as previously described (Jacquin *et al*, 2017). Briefly, cells were fixed with ice-cold methanol at −20°C for 5 min. Cells were blocked in 5% BSA in PBS for 1 h at room temperature and incubated in primary antibody in blocking buffer overnight at 4°C. Following washing, cells were stained with appropriate secondary antibodies and DNA stained by DAPI before being mounted using Prolong Gold Antifade (P36934, Thermofisher). Image acquisition was made using a Zeiss LSM 780 confocal microscope (Carl Zeiss Ltd), using Zen software (Carl Zeiss Ltd).

## Latex bead LC3-associated phagocytosis assay

Cells were plated on glass coverslips and the following day, 3-micron uncoated polystyrene beads (672326, Polysciences) were added for 4 h before addition of 100 μM monensin for 1 h. The cells were washed with PBS and fixed with methanol at −20°C for 5 min before processing for immunofluorescence.

## Apoptotic cell LC3-associated phagocytosis assay

MEF cells were plated on 35-mm glass-bottomed dishes (MatTek). Apoptotic corpses were prepared by UV crosslinking CellTracker Red (Invitrogen, C34552) stained HCT116 *ATG16L1*$^{-/-}$ cells with two rounds of 8,000 μJ using a UV stratalinker 2400 (Stratagene). Corpses were then added to MEF cells at a ratio of 5:1. After 14 h, cells were imaged live using a confocal Zeiss LSM 780 microscope (Carl Zeiss Ltd) equipped with a 63× oil immersion objective. The presence of GFP-LC3 on 20 apoptotic cell-containing phagosomes was quantified per condition per experiment.

## Macropinocytosis assay

GFP-LC3-expressing MEF cells were plated on 35-mm glass-bottomed dishes (MatTek). The next day cells were serum starved

for 24 h followed by stimulation with media containing 0.1 μg/ml PDGF and 0.1 mg/ml tetramethylrhodamine conjugated dextran (fluoro-Ruby; D-1817 Life Technologies). Live microscopy was performed in an incubation chamber at 37°C, with 5% $CO_2$. Images of newly formed, dextran-containing macropinosomes, were acquired in live cells using a confocal Zeiss LSM 780 microscope (Carl Zeiss Ltd) equipped with a 40× oil immersion 1.4 NA objective, using Zen software (Carl Zeiss Ltd).

## Entotic Corpse assay

MCF10A GFP-LC3 cells were plated on glass coverslips overnight to allow cell-in-cell structures to form and mature, followed by treatment with 100 μM monensin for 1 h. Cells were washed and processed for immunofluorescence. LAMP1-positive corpse-containing vacuoles were analysed for GFP-LC3 signal.

## VacA toxin assay

MEF cells were plated on 35-mm glass-bottomed dishes (MatTek) and treated with VacA toxin (10 μM; kindly provided by Dr. Timothy Cover) as previously described (Florey *et al*, 2015).

## Bone marrow-derived dendritic cell (BMDC) isolation

C57/BL6 wild-type and ATG16L1 E230 mice, aged 13–15 weeks, were used to obtain BMDCs. A neomycin-targeting vector for Atg16l1 was generated with two stop codons introduced into exon 6 after amino acid position 230. R1 ES cells were electroporated with linearised targeting vector and G418 resistant clones screened by Southern blot and PCR. Positive clones were injected into C57/B6 blastocysts. Chimeric founder mice were crossed with C57/B6 females and the neomycin cassette removed by crossing F1 offspring with FlpO transgenic mice and then crossed with a C57/B6 background. The E230 mice are available from Drs Ulrike Mayer and Thomas Wileman. Bone marrow cells were isolated by flushing tibias and femurs with PBS + 2% FBS. Cells were pelleted and resuspended in 1 ml Red Blood Cell lysis buffer (150 mM $NH_4Cl$, 10 mM $KHCO_3$, 0.1 mM EDTA) for 2 min at room temperature. Cells were pelleted and resuspended in RPMI 1640 (Invitrogen 22409-031), 10% FBS, 1% Pen/Strep, 50 μM 2-mercaptoethanol supplemented with 20 ng/ml murine GM-CSF (Peprotech, 315-03), 10 ng/ml murine IL-4 (Peprotech, AF-214-14) and 50 ng/ml Fungizone (Amphotericin B; Gibco, 15290018). Media was refreshed on days 3 and 6 and used for antigen presentation assays on day 8.

## Antigen presentation assay

$2.5 × 10^5$ *in vitro* differentiated BMDCs were plated in 24-well plates and incubated for 24 h with different concentrations of Eα-GFP antigen (kindly provided by Dr. Michelle Linterman, Babraham Institute). Cells were collected and incubated for 10 min at 4°C for with FC blocking media (eBioscience, 14-0161-82), before staining with biotin-labelled Y-Ae antibody (eBioscience, 13-5741-85), which recognises the Eα-MHCII complex, for 1 h at 4°C. Cells were washed before incubation with PE-labelled anti-CD11b (clone M1/70, BD Horizon, 562287), APC-eFluor780-labelled anti-CD11c (clone N418,

eBioscience, 47-0114-82), Alexa Fluor 700-labelled anti-MHC class II (clone M5/114.15.2, eBioscience, 56-5321-82) and PE-labelled strep-tavidin (eBioscience, 12-4317-87) for 1 h at 4°C. Cells were analysed by flow cytometry using a Fortessa A (Becton Dickinson) machine and data analysed using FlowJo software.

### Statistics

Data were analysed by a two-tailed unpaired Student's *t*-test, using Prism 6 software.

**Expanded View** for this article is available online.

### Acknowledgements

We thank Felix Randow for helpful comments and suggestions, and David Oxley in the Babraham Institute Mass Spectrometry facility, and the Babraham imaging facility. This research was supported by the Cambridge NIHR BRC Cell Phenotyping Hub. This work was funded by Cancer Research UK (C47718/A16337, O.F.), the Medical Research Council (RG89611, R.B.) and the BBSRC Institute Strategic Programme Gut Health and Food Safety (BB/J004529/1).

### Author contributions

KF and RU designed performed and analysed experiments. EJ and TV performed and analysed experiments. NG provided essential reagents and reviewed the manuscript. UM and JMA designed and made the E230 mouse. SC provided financial and intellectual input. TW designed experiments. RB and OF designed experiments and wrote the paper.

### Conflict of interest

The authors declare that they have no conflict of interest.

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
