## [Review Process File · The EMBO Journal]

The WD40 domain of ATG16L1 is required for its non-canonical role in lipidation of LC3 at single membranes

Katherine Fletcher, Rachel Ulferts, Elise Jacquin, Talitha Veith, Noor Gammoh, Julia M. Arasteh, Ulrike Mayer, Simon R. Carding, Thomas Wileman, Rupert Beale & Oliver Florey

Review timeline:

Submission date:	24 July 2017
Editorial Decision:	9 August 2017
Revision received:	10 November 2017
Editorial Decision:	22 November 2017
Revision received:	6 December 2017
Accepted:	14 December 2017

Editor: Andrea Leibfried

Transaction Report:

1st Editorial Decision

9 August 2017

Thank you for submitting your manuscript for consideration by the EMBO Journal. It has now been seen by three referees whose comments are shown below.

As you will see, the referees appreciate your data. However, they also think that the study needs to be extended to warrant publication in The EMBO Journal.

Should you be able to extend your work as outlined by the referees, I would like to invite you to submit a revised version of the manuscript. Importantly, the revision should provide strong data to offer:

- further reaching insight into how the WD40 domain mediates recruitment of ATG16L1 (see also report from referee #2), and this is also crucial to clearly elevate your findings above the work from Boada-Romero et al, 2016
- insight into the functional consequences of WD40 mediated ATG16L1 activity (see also report from referee #3).

All other points raised by the referees should also be addressed. Should you not be able to address the comments of all three reviewers, please get in touch with me. Please also contact me in case you would like to discuss the revision further. I should add that it is EMBO Journal policy to allow only a single round of revision, and acceptance of your manuscript will therefore depend on the completeness of your responses in this revised version.

Thank you for the opportunity to consider your work for publication. I look forward to your revision.

REFeree REPORTS

Referee #1:

Fletcher et al. investigated the role of ATG16L1 in what they term "non-canonical autophagy", which is defined by single membrane endolysosomal vesicles labeled with LC3. While ATG16L1 WD repeat containing C-terminal domain (WD40 CTD) is dispensable for canonical autophagy the authors provide evidence that this region of the protein is essential for targeting to single-membrane vesicles and consequently promotion of LC3 lipidation. Employing truncated ATG16L1, the authors successfully distinguished between canonical autophagy that requires VPS34 and WIPI2b and non-canonical autophagy defined by ATG16L1 recruitment and LC3 lipidation independently of PI3P and WIPI2b. The requirement of WD40 CTD is clearly demonstrated under physiological conditions such as LC3-associated phagocytosis (LAP), exposure to the bacterial toxin VacA and infection by influenza A virus.

This research provides a solid and straightforward data evaluating a specific role for WD40 CTD of ATG16L1. The authors also provided a genetic tool that clearly enables to distinguish between macroautophagy and non-canonical usage of autophagic machinery. These findings contribute to better understanding of cellular pathways utilizing parts of the autophagic system.

Specific comments

1. The effect of the sodium/proton ionophore monensin described by the authors in this and in their previous study is rather unclear. Accordingly, the fact that it leads to LC3 lipidation may be explained not only by induction of osmotic imbalances within endosomal compartments, but an inhibition of ATG4 de-lipidation activity, which in principle will lead to increased LC3 lipidation. This should be better addressed textually and experimentally.
2. Figure 2e - high LC3 lipidation in Δ FBD mutant is not in correlation with that data presented in Figure 2g for this mutant under starvation.
3. Figure 2e and Figure 3a - no explanation is given for the elevated LC3 lipidation in the control of Δ FBD mutant.
4. The term non-canonical autophagy in this context is somewhat confusion and the authors are encouraged to better define this process.

Referee #2:

The manuscript by Fletcher et al. reports the interesting finding that the C-terminal WD40 domain of human ATG16L1 is required for macroautophagy independent lipidation of LC3.

ATG16L1 is part of the ATG5-12-16 complex, which acts in a E3-like manner to promote the conjugation of LC3 proteins to the membrane lipid PE. This process is canonically associated with macroautophagy, where cells conjugate LC3 proteins to the nascent autophagosomal membrane. However, it has become evident that LC3 proteins are also lipidated in processes that do not involve double membranes (such as the autophagosome). The ATG16L1 protein is required for all these LC3 lipidation events and the authors here show, using transduced cell lines, that the C-terminal WD40 domain of ATG16L1 is required for macroautophagy independent LC3 lipidation but not for lipidation during macroautophagy. Mechanistic insights into how the WD40 domain mediates the recruitment of ATG16L1 in the macroautophagy independent processes are not provided. Does it bind lipids, other proteins or both? In my opinion, the manuscript will be of interest for the autophagy community but in its current form it is too limited in its scope to be a strong candidate for the EMBO Journal.

Specific comments

1. I think the term non-canonical autophagy is misleading and not ideal to describe the processes studied in the paper. In reality, these are not autophagic processes as no "self" is degraded. Instead, most processes studied in the manuscript are phagocytic or endocytic in nature. For a detailed discussion please see for example a recent review in the EMBOJ (Galluzzi, 2017, EMBOJ (PMID: 28596378)).
2. Are there other assays than LC3 lipidation to assess "non-canonical" autophagy? For example, the delivery of material into lysosomes and/or its degradation? Are the processes studied actually dependent on LC3 lipidation and blocked by deletion of the WD40 domain of ATG16L1?

3. In Figure 2e, was Bafilomycin used? Also, the quantification does not accurately reflect what can be seen in the blot (Figure 2e, f).

4. The statement "The structure of the WD40 CTD has recently been solved, but its biological function remains unclear 20." is not entirely correct (see Boada-Romero, 2016, Nat Comms (PMID: 27273576)).

Referee #3:

Autophagy is characterized by the lipidation of the ubiquitin-like molecule LC3, which is mediated by the ATG16L1 E3 ligase-like complex. This event contributes to the formation of the double-membrane autophagosome. Recent studies indicate that lipidated LC3 can also insert into single membrane vesicles to mediate autophagy-related processes such as LC3-associated phagocytosis (LAP). The mechanisms that distinguish these disparate functions of LC3 are incompletely understood. Previous studies have shown that the WD40 domain of ATG16L1 is dispensable for conventional starvation-induced autophagy. In this manuscript, the authors examine whether the WD40 domain is necessary for other functions associated with LC3 lipidation by ATG16L1.

The authors use LAP and monensin-induced LC3 targeting as models of "non-canonical autophagy". They first show that recruitment of LC3 and/or ATG16L1 to single membranes occurs independent of VPS34 and WIPI2b, factors typically required for targeting of LC3 to double membranes during autophagy. Consistent with this finding, they show that the ATG16L1 truncation mutant lacking the WIPI2b and FIP200 binding region (delta FBD) can mediate LC3 targeting to single membranes when introduced into cell lines in which endogenous ATG16L1 is removed by CRISPR/Cas9. In contrast, ATG16L1 lacking the WD40 domain (delta WD40) are deficient in LC3 targeting to single membranes, but retain the ability to mediate autophagy. The inability to mediate LC3 targeting to single membranes was not due to failure to form the E3 ligase-like complex (ATG5-ATG12-ATG16L1), but was associated with lack of recruitment of ATG16L1 delta WD40 to the phagosome. Finally, they show that the same pathway is triggered by influenza M2 protein, which inserts into membranes to act as a proton channel and can have a similar effect as monensin. In a manner dependent on the proton channel activity of M2, influenza infection induces relocalization of LC3 in cells harboring ATG16L1 delta FBD but not ATG16L1 delta WD40.

The role of the WD40 domain in mediating LC3 targeting is novel and the observation that autophagy and this autophagy-related function of ATG16L1 can be decoupled through mutagenesis is exciting. The experiments include validation through multiple techniques with appropriate quantification and controls. The use of several cell lines and results with influenza infection are additional strengths. Thus, the conclusions are generally supported well by the data. The manuscript could be strengthened further if the authors can address the following concerns:

1. The terminology used in this manuscript is confusing. As articulated in a recent comprehensive review article (Galluzi et al, EMBO J. 2017 36(13):1811), the term "non-canonical autophagy" is ambiguous and misleading.
2. A related issue is that the relationship between LAP, monensin treatment, and M2 activity is obscure. Are these the same events at the molecular level? For instance, does LC3 targeting to single membranes in the presence of monensin or M2 require rubicon (an essential LAP protein)? Unifying these disparate models would significantly improve this manuscript and allow the authors to use the umbrella term LAP. It would also justify switching between models. Alternatively, in some experiments it appears that monensin enhances LAP. Is this enhancement or induction?
3. Most experiments rely on transformed cell lines, which typically display dysregulated autophagy. The exception is the MEFs. Are the ATG16L1 knockout MEFs transformed? Information on passage number would be helpful.
4. Although LC3 lipidation and trafficking is clearly affected by ATG16L1 mutagenesis, the authors do not provide evidence demonstrating a functional consequence. For instance, LAP has clear

functions in pathogen control and cytokine production. Are these dependent on the WD40 domain? In the example of influenza M2, does the WD40 domain affect viral replication or immune responses to the virus? The authors refer to controversies in the field regarding the relationship between M2 and autophagy. They appear to have an opportunity to address this through functional studies. The authors are not expected to chase every downstream function of LC3 targeting that could be important, but certainly some evidence should be provided to show that the WD40 domain matters beyond LC3 Western blots and immuno-fluorescence.

1st Revision - authors' response

10 November 2017

Referee #1:

Fletcher et al. investigated the role of ATG16L1 in what they term "non-canonical autophagy", which is defined by single membrane endolysosomal vesicles labeled with LC3. While ATG16L1 WD repeat containing C-terminal domain (WD40 CTD) is dispensable for canonical autophagy the authors provide evidence that this region of the protein is essential for targeting to single-membrane vesicles and consequently promotion of LC3 lipidation. Employing truncated ATG16L1, the authors successfully distinguished between canonical autophagy that requires VPS34 and WIPI2b and non-canonical autophagy defined by ATG16L1 recruitment and LC3 lipidation independently of PI3P and WIPI2b. The requirement of WD40 CTD is clearly demonstrated under physiological conditions such as LC3-associated phagocytosis (LAP), exposure to the bacterial toxin VacA and infection by influenza A virus.

This research provides a solid and straightforward data evaluating a specific role for WD40 CTD of ATG16L1. The authors also provided a genetic tool that clearly enables to distinguish between macroautophagy and non-canonical usage of autophagic machinery. These findings contribute to better understanding of cellular pathways utilizing parts of the autophagic system.

We thank the reviewer for their positive comments.

Specific comments

1. The effect of the sodium/proton ionophore monensin described by the authors in this and in their previous study is rather unclear. Accordingly, the fact that it leads to LC3 lipidation may be explained not only by induction of osmotic imbalances within endosomal compartments, but an inhibition of ATG4 de-lipidation activity, which in principle will lead to increased LC3 lipidation. This should be better addressed textually and experimentally.

We understand the reviewer's point, that revealing LC3 lipidation could in principle be due to induction of lipidation or prevention of ATG4 de-lipidation. However we do not believe the LC3 lipidation we observe is related to inhibition of ATG4 activity. We are able to induce endolysosomal LC3 lipidation with a wide variety of lysosomotropic drugs in addition to monensin (Florey et al, 2015; Jacquin et al, 2017). Indeed, we are able to induce lipidation simply by altering the osmotic properties of the media. Under these osmotic conditions LC3 is only seen to lipidate to single-membrane endolysosomal compartments rather than early endosomes or the plasma membrane. If there was an inhibition of ATG4 activity we may expect to see indiscriminate LC3 relocalization to membranes. We have now included extra experimental FRAP data (Fig EV2 C and D), which shows that monensin treatment induces the prolonged recruitment of ATG16L1 to lysosomal membranes, which we believe is the driver of LC3 lipidation under this system.

2. Figure 2e - high LC3 lipidation in Δ FBD mutant is not in correlation with that data presented in Figure 2g for this mutant under starvation.

Data from Figure 2E is from HCT116 cells, while 2G come from MEF cells. The corresponding confocal images for HCT116 cells are found in Fig EV 1A.

3. Figure 2e and Figure 3a - no explanation is given for the elevated LC3 lipidation in the control of Δ FBD mutant.

It has previously been shown (Gammoh et al, 2013) that deletion of the FBD does not eliminate basal autophagy, or glucose starvation induced autophagy. Hence, we would not expect to see a complete loss of LC3-II in these cell lines. The quantification data shown was obtained from blots from three independent experiments, which – like all biological samples- showed some variability. The blot shown here is one of these blots.

4. The term non-canonical autophagy in this context is somewhat confusion and the authors are encouraged to better define this process.

We thank the reviewers for raising this interesting point. We agree that the term “non-canonical” autophagy is not perfect, as it does not fit within narrow definitions of what can be termed an autophagic process. However, there exists a growing body of work in which the term non-canonical autophagy has already been applied to the lipidation of LC3 to single-membrane compartments (Henault et al, 2012; Kim et al, 2013; Martinez et al, 2016). Indeed, many reviews and guidelines on autophagy have termed the process “non-canonical autophagy” or a “noncanonical autophagy process” (Cadwell, 2016; Klionsky et al, 2016). LC3 associated phagocytosis, or LAP, is commonly used to describe these events. However, many of the processes we study are not phagocytosis, and there are some molecular mechanisms that appear to be specific to phagocytosis rather than the non-canonical autophagy process in general (see response to reviewer 3 point 2). We considered introducing another term to describe LC3 lipidation at single membranes that is dependent on the WD40 domain of ATG16L1, but decided that yet another acronym to describe this set of processes would if anything lead to more confusion. We have therefore altered our text in the introduction to make this point as clear as possible, and define precisely what we mean when we use the term non-canonical autophagy (pages 4-5). We have also altered the title to make it clearer on what processes we are studying.

Referee #2:

The manuscript by Fletcher et al. reports the interesting finding that the C-terminal WD40 domain of human ATG16L1 is required for macroautophagy independent lipidation of LC3.

ATG16L1 is part of the ATG5-12-16 complex, which acts in a E3-like manner to promote the conjugation of LC3 proteins to the membrane lipid PE. This process is canonically associated with macroautophagy, where cells conjugate LC3 proteins to the nascent autophagosomal membrane. However,

it has become evident that LC3 proteins are also lipidated in processes that do not involve double membranes (such as the autophagosome). The ATG16L1 protein is required for all these LC3 lipidation events and the authors here show, using transduced cell lines, that the C-terminal WD40 domain of ATG16L1 is required for macroautophagy independent LC3 lipidation but not for lipidation during macroautophagy. Mechanistic insights into how the WD40 domain mediates the recruitment of ATG16L1 in the macroautophagy independent processes are not provided. Does it bind lipids, other proteins or both? In my opinion, the manuscript will be of interest for the autophagy community but in its current form it is too limited in its scope to be a strong candidate for the EMBO Journal.

We thank the reviewer for their positive comments. We have included new data (Fig 6) to increase our mechanistic understanding how ATG16L1 functions during non-canonical autophagy. We now identify key amino acid sites within the WD40 CTD that are required for non-canonical autophagy. This is the first report of such sites and increases our understanding of how the WD40 CTD orchestrates LC3 lipidation.

Specific comments

1. I think the term non-canonical autophagy is misleading and not ideal to describe the processes studied in the paper. In reality, these are not autophagic processes as no "self" is degraded. Instead, most processes studied in the manuscript are phagocytic or endocytic in nature. For a detailed discussion please see for example a recent review in the EMBOJ (Galluzzi, 2017, EMBOJ (PMID: 28596378)).

We thank the reviewer for this comment and refer them to our answer to referee 1 point 4.

2. Are there other assays than LC3 lipidation to assess "non-canonical" autophagy? For example, the delivery of material into lysosomes and/or its degradation? Are the processes studied actually dependent on LC3 lipidation and blocked by deletion of the WD40 domain of ATG16L1?

We thank the reviewer for this suggestion. The functions of non-canonical autophagy are still being determined. However, a number of immune cell functions have been proposed to be dependent on it. We have utilized a new mouse model where the C-terminal WD40 domain of ATG16L1 has been truncated. This renders the mouse deficient for non-canonical autophagy, while remaining competent for canonical autophagy. Using this system we now demonstrate a requirement for non-canonical autophagy in dendritic cell MHC class II antigen presentation (Fig 7).

3. In Figure 2e, was Bafilomycin used? Also, the quantification does not accurately reflect what can be seen in the blot (Figure 2e, f).

Bafilomycin was not included in the experiment. For the quantification, blots of three independently experiments were used. Only one of these

blots can unfortunately be included in the paper. The main point of the experiment was to demonstrate that the Δ FBD cells show a defect in canonical autophagy responses to mTor inhibition, while the Δ WD cell do not. We have adjusted the analysis to show fold induction of LC3II/LC3I ratios over control (Fig 2 F). This shows that Δ FBD cells respond significantly less to PP242 than full length expressing cells, where as there is no difference between FL and Δ WD cells. These western blots are just one assay to test the autophagic response, as we have also performed LC3 and WIPI2b puncta counts. All data support our conclusions, and these are in line with previous publications that show the same result.

4. The statement "The structure of the WD40 CTD has recently been solved, but its biological function remains unclear 20." is not entirely correct (see Boada-Romero, 2016, Nat Comms (PMID: 27273576)).

We agree with the reviewer that recent work has demonstrated some role for the WD40 CTD and have altered our text accordingly to reference this, Page 7 line 131.

Further to this we now include data, which identifies key residues within the WD40 CTD that are required for non-canonical autophagy (Fig 6). This is the first report of residues within the WD40 CTD of ATG16L1 that affect its function. This further distinguishes our work from that of Boada-Romero et al, and extends our understanding of how the WD40 CTD controls LC3 lipidation to single membranes.

Referee #3:

Autophagy is characterized by the lipidation of the ubiquitin-like molecule LC3, which is mediated by the ATG16L1 E3 ligase-like complex. This event contributes to the formation of the double-membrane autophagosome. Recent studies indicate that lipidated LC3 can also insert into single membrane vesicles to mediate autophagy-related processes such as LC3-associated phagocytosis (LAP). The mechanisms that distinguish these disparate functions of LC3 are incompletely understood. Previous studies have shown that the WD40 domain of ATG16L1 is dispensable for conventional starvation-induced autophagy. In this manuscript, the authors examine whether the WD40 domain is necessary for other functions associated with LC3 lipidation by ATG16L1.

The authors use LAP and monensin-induced LC3 targeting as models of "non-canonical autophagy". They first show that recruitment of LC3 and/or ATG16L1 to single membranes occurs independent of VPS34 and WIPI2b, factors typically required for targeting of LC3 to double membranes during autophagy. Consistent with this finding, they show that the ATG16L1 truncation mutant lacking the WIPI2b and FIP200 binding region (Δ FBD) can mediate LC3 targeting to single membranes when introduced into cell lines in which endogenous ATG16L1 is removed by CRISPR/Cas9. In contrast, ATG16L1 lacking the WD40 domain (Δ WD40) are deficient in

LC3 targeting to single membranes, but retain the ability to mediate autophagy. The inability to mediate LC3 targeting to single membranes was not due to failure to form the E3 ligase-like complex (ATG5-ATG12-ATG16L1), but was associated with lack of recruitment of ATG16L1 delta WD40 to the phagosome. Finally, they show that the same pathway is triggered by influenza M2 protein, which inserts into membranes to act as a proton channel and can have a similar effect as monensin. In a manner dependent on the proton channel activity of M2, influenza infection induces relocalization of LC3 in cells harboring ATG16L1 delta FBD but not ATG16L1 delta WD40.

The role of the WD40 domain in mediating LC3 targeting is novel and the observation that autophagy and this autophagy-related function of ATG16L1 can be decoupled through mutagenesis is exciting. The experiments include validation through multiple techniques with appropriate quantification and controls. The use of several cell lines and results with influenza infection are additional strengths. Thus, the conclusions are generally supported well by the data. The manuscript could be strengthened further if the authors can address the following concerns:

We thank the reviewer for their encouraging comments. We have performed new experiments to address the questions regarding the consequences of non-canonical autophagy.

1. The terminology used in this manuscript is confusing. As articulated in a recent comprehensive review article (Galluzi et al, EMBO J. 2017 36(13):1811), the term "non-canonical autophagy" is ambiguous and misleading.

We than the reviewer for this comment and refer them to our response to referee 1 point 4.

2. A related issue is that the relationship between LAP, monensin treatment, and M2 activity is obscure. Are these the same events at the molecular level? For instance, does LC3 targeting to single membranes in the presence of monensin or M2 require rubicon (an essential LAP protein)? Unifying these disparate models would significantly improve this manuscript and allow the authors to use the umbrella term LAP. It would also justify switching between models. Alternatively, in some experiments it appears that monensin enhances LAP. Is this enhancement or induction?

This is an interesting point which requires clarification within the field. Data from our laboratories show that all tested processes that activate non-canonical autophagy (phagocytosis, macropinocytosis, entosis, influenza infection, ionophore and lysosomotropic drugs) require V-ATPase activity and the LC3 lipidation machinery – ATG5, ATG7, ATG12 and ATG16L1 (specifically now the WD40 CTD). The requirement for Rubicon and ROS appears to be more specific for LAP, rather than being essential for non-canonical autophagy in general. Indeed, in unpublished work we are able to induce LC3 lipidation to phagosomes from cells lacking Rubicon, or where NADPH oxidase has been

inhibited, by treating the cells with monensin. It is possible that during LAP, Rubicon and ROS act to generate another “signal” that activates non-canonical autophagy. This “signal” may be generated through different mechanisms in other processes. As such we do not believe LAP can be used as an umbrella term.

3. Most experiments rely on transformed cell lines, which typically display dysregulated autophagy. The exception is the MEFs. Are the ATG16L1 knockout MEFs transformed? Information on passage number would be helpful.

We share the reviewer’s concern – indeed we decided to demonstrate the same dependence on the WD40 CTD in three separate knock-out cell lines to avoid drawing conclusions that might be specific to dysregulated autophagy peculiar to a particular cell type. To exclude the possibility that all cell lines tested are similarly dysregulated we now include data from primary mouse dendritic cells that lack the WD40 CTD (Fig 7).

4. Although LC3 lipidation and trafficking is clearly affected by ATG16L1 mutagenesis, the authors do not provide evidence demonstrating a functional consequence. For instance, LAP has clear functions in pathogen control and cytokine production. Are these dependent on the WD40 domain? In the example of influenza M2, does the WD40 domain affect viral replication or immune responses to the virus? The authors refer to controversies in the field regarding the relationship between M2 and autophagy. They appear to have an opportunity to address this through functional studies. The authors are not expected to chase every downstream function of LC3 targeting that could be important, but certainly some evidence should be provided to show that the WD40 domain matters beyond LC3 Western blots and immuno-fluorescence.

This is an important point. What are the functions of non-canonical autophagy? We very much appreciate the reviewer’s comment that we are not expected to chase every downstream function of LC3 targeting. To address the issue of functional consequences of inhibiting non-canonical autophagy through deletion of the WD40 CTD, we used primary dendritic cells generated from a mouse which lacks the C-terminal domain of ATG16L1. We show that presentation of exogenous antigen on MHC class II is deficient in cells from this mouse, demonstrating an important functional role for non-canonical autophagy dependent on ATG16L1 WD40 CTD in a non-transformed cell (Fig 7).

To exclude the possibility that the effect of ATG16L1 WD40 CTD on non-canonical autophagy during influenza infection is due to a replication deficiency, we measured the kinetics of influenza infection in ATG16L1 deficient HCT116 cells and found no difference between uncomplemented or FL and Δ WD expressing cells (Fig EV4). Cell lines do not model the whole life cycle of influenza and laboratory adapted influenza strains for which reverse genetic systems are readily available (such as PR8) are not efficient inducers of cell-autonomous immune responses in cell culture. It is likely there will be important functional

consequences for influenza infection in vivo as the virus encodes an evolutionarily conserved LC3 interacting region: it would be very interesting to study non-canonical autophagy and immune responses to influenza in a whole organism context (though mice are not ideal for such studies, a ferret model is preferred). These therefore are important questions which go beyond the scope of this paper. We believe that our demonstration of the importance of the WD40 domain of ATG16L1 will allow such questions to be interrogated with much greater precision in the future.

References

- Cadwell K (2016) Crosstalk between autophagy and inflammatory signalling pathways: balancing defence and homeostasis. *Nat Rev Immunol* **16**: 661-675
- Florey O, Gammoh N, Kim SE, Jiang X, Overholtzer M (2015) V-ATPase and osmotic imbalances activate endolysosomal LC3 lipidation. *Autophagy* **11**: 88-99
- Gammoh N, Florey O, Overholtzer M, Jiang X (2013) Interaction between FIP200 and ATG16L1 distinguishes ULK1 complex-dependent and-independent autophagy. *Nature structural & molecular biology* **20**: 144-149
- Henault J, Martinez J, Riggs JM, Tian J, Mehta P, Clarke L, Sasai M, Latz E, Brinkmann MM, Iwasaki A, Coyle AJ, Kolbeck R, Green DR, Sanjuan MA (2012) Noncanonical autophagy is required for type I interferon secretion in response to DNA-immune complexes. *Immunity* **37**: 986-997
- Jacquin E, Leclerc-Mercier S, Judon C, Blanchard E, Fraitag S, Florey O (2017) Pharmacological modulators of autophagy activate a parallel noncanonical pathway driving unconventional LC3 lipidation. *Autophagy*: 1-14
- Kim JY, Zhao H, Martinez J, Doggett TA, Kolesnikov AV, Tang PH, Ablonczy Z, Chan CC, Zhou Z, Green DR, Ferguson TA (2013) Noncanonical autophagy promotes the visual cycle. *Cell* **154**: 365-376
- Klionsky DJ, Abdelmohsen K, Abe A, Abedin MJ, Abeliovich H, Acevedo Arozena A, Adachi H, Adams CM, Adams PD, Adeli K (2016) Guidelines for the use and interpretation of assays for monitoring autophagy. *Autophagy* **12**: 1-222
- Martinez J, Cunha LD, Park S, Yang M, Lu Q, Orchard R, Li QZ, Yan M, Janke L, Guy C, Linkermann A, Virgin HW, Green DR (2016) Noncanonical autophagy inhibits the autoinflammatory, lupus-like response to dying cells. *Nature* **533**: 115-111

2nd Editorial Decision

22 November 2017

Thank you for submitting your manuscript for consideration by the EMBO Journal. It has now been seen by the three original referees again whose comments are enclosed. As you will see, all three referees express interest in your manuscript and are broadly in favour of publication, pending satisfactory minor revision.

I would thus like to ask you to address referee #2 and #3's remaining concerns and to provide a final version of your manuscript.

Thank you for the opportunity to consider your work for publication. I look forward to your revision.

REFEREE REPORTS

Referee #1:

The authors addressed my concerns and the manuscript now meets EMBO J scientific merit.

Referee #2:

The authors have addressed all my comments and have added substantial mechanistic insights. In my opinion, the manuscript has become very strong. One thing the authors should still do is to add a loading control for the blot shown in Figure 6D.

Referee #3:

In this revised manuscript, the authors provide new data demonstrating that the WD40 domain of ATG16L1 mediates autophagy-related processes that are distinct from starvation-induced autophagy. The authors were generally responsive to previous critiques, and the manuscript is much improved. In particular, they include key data with mice deficient in the WD40 domain (ATG16L1 E230 mice) demonstrating physiological relevance of their findings. Their observations are consistent with the literature indicating that LAP or a similar pathway is necessary for presentation of exogenous antigens by dendritic cells.

The authors should include information on the origin of the ATG16L1 E230 mice and how they were generated. The manuscript is otherwise appropriate for publication.

2nd Revision - authors' response

6 December 2017

Referee #2:

The authors have addressed all my comments and have added substantial mechanistic insights. In my opinion, the manuscript has become very strong. One thing the authors should still do is to add a loading control for the blot shown in Figure 6D.

We have now included a loading control for figure 6D.

Referee #3:

In this revised manuscript, the authors provide new data demonstrating that the WD40 domain of ATG16L1 mediates autophagy-related processes that are distinct from starvation-induced autophagy. The authors were generally responsive to previous critiques, and the manuscript is much improved. In particular, they include key data with mice deficient in the WD40 domain (ATG16L1 E230 mice) demonstrating physiological relevance of their findings. Their observations are

consistent with the literature indicating that LAP or a similar pathway is necessary for presentation of exogenous antigens by dendritic cells.

The authors should include information on the origin of the ATG16L1 E230 mice and how they were generated. The manuscript is otherwise appropriate for publication.

As we stated in our previous point-by point response, the generation of this mouse is being published in another manuscript from the labs of Dr Thomas Wileman and Dr Ulrike Mayer. We added basic information on the how the ATG16L1 gene was targeted. We hope the reviewer understands why we cannot provide more information at this time.

Corresponding Author Name: Oliver Florey

Manuscript Number: EMBOJ-2017-97840R